# Molecular definition of multiple sites of antibody inhibition of malaria transmission-blocking vaccine antigen Pfs25

Stephen W. Scally[1], Brandon McLeod[1,2], Alexandre Bosch[1], Kazutoyo Miura[3], Qi Liang[4], Sean Carroll[5], Sini Reponen[5], Ngan Nguyen[5], Eldar Giladi[5], Sebastian Rämisch [6], Vidadi Yusibov[7], Allan Bradley[4,8], Franck Lemiale[9], William R. Schief[6], Daniel Emerling[5], Paul Kellam[4,10], C. Richter King[9] & Jean-Philippe Julien[1,2,11]

The *Plasmodium falciparum* Pfs25 protein (Pfs25) is a leading malaria transmission-blocking vaccine antigen. Pfs25 vaccination is intended to elicit antibodies that inhibit parasite development when ingested by *Anopheles* mosquitoes during blood meals. The Pfs25 three-dimensional structure has remained elusive, hampering a molecular understanding of its function and limiting immunogen design. We report six crystal structures of Pfs25 in complex with antibodies elicited by immunization via Pfs25 virus-like particles in human immunoglobulin loci transgenic mice. Our structural findings reveal the fine specificities associated with two distinct immunogenic sites on Pfs25. Importantly, one of these sites broadly overlaps with the epitope of the well-known 4B7 mouse antibody, which can be targeted simultaneously by antibodies that target a non-overlapping site to additively increase parasite inhibition. Our molecular characterization of inhibitory antibodies informs on the natural disposition of Pfs25 on the surface of ookinetes and provides the structural blueprints to design next-generation immunogens.

[1] Program in Molecular Medicine, The Hospital for Sick Children Research Institute, 686 Bay St, Toronto, ON, Canada M5G 0A4. [2] Department of Biochemistry, University of Toronto, 1 King's College Circle, Toronto, ON, Canada M5S 1A8. [3] Laboratory of Malaria and Vector Research, National Institute of Allergy and Infectious Diseases, National Institutes of Health, 12735 Twinbrook Parkway, Rockville, MD 20852, USA. [4] Kymab Ltd., The Bennet Building (B930) Babraham Research Campus, Cambridge, CB22 3AT, UK. [5] Atreca, 500 Saginaw Drive, Redwood City, CA 94063-4750, USA. [6] Department of Immunology and Microbiology, The Scripps Research Institute, La Jolla, CA 92037, USA. [7] Fraunhofer USA Center for Molecular Biotechnology CMB, 9 Innovation Way, Newark, DE 19711, USA. [8] Wellcome Trust Sanger Institute, Hinxton, Cambridge CB10 1SA, UK. [9] PATH's Malaria Vaccine Initiative, 455 Massachusetts Avenue NW Suite 1000, Washington, DC 20001, USA. [10] Department of Medicine, Division of Infectious Diseases, Imperial College London, London SW7 2AZ, UK. [11] Department of Immunology, University of Toronto, 1 King's College Circle, Toronto, ON, Canada M5S 1A8. Correspondence and requests for materials should be addressed to J.-P.J. (email: jean-philippe.julien@sickkids.ca)

A major challenge for vaccine development against *Plasmodium falciparum* (*Pf*) is its complex lifecycle involving both an asexual stage in the human host, and a sexual stage primarily in the *Anopheles* mosquito. Generally, it is assumed that the most effective path to malaria elimination will involve a combination of immunization strategies effective at blocking several *Pf* life stages[1,2]. Fundamental to this idea are transmission-blocking vaccines (TBVs), which seek to inhibit the development of parasites in the mosquito vector to prevent its spread back to the human population. It has been well documented that antibody interference with specific sexual stage antigens can lead to the inhibition of *Pf* in *Anopheles* mosquitoes[3–5].

The TBV candidate antigen Pfs25, which has been tested extensively in human trials[6,7], is a glycosylphosphatidylinositol-linked protein expressed on the surface of ookinetes[8,9]. Pfs25 is important for ookinete survival in the protease-rich mosquito midgut, assists with penetration of the mosquito epithelium, and aids maturation of ookinetes into oocysts[10,11]. Pfs25 is predicted to fold into four EGF-like domains and to contain multiple internal disulfide bonds, but its three-dimensional structure has not yet been solved[8,12]. Much of our structural understanding of Pfs25 is based on a model derived from the homologous protein expressed by *Plasmodium vivax* (Pvs25, 46% sequence identity) for which the atomic structure is known[13]. Pfs25 protein is expressed solely in the mosquito, and consequently, its low

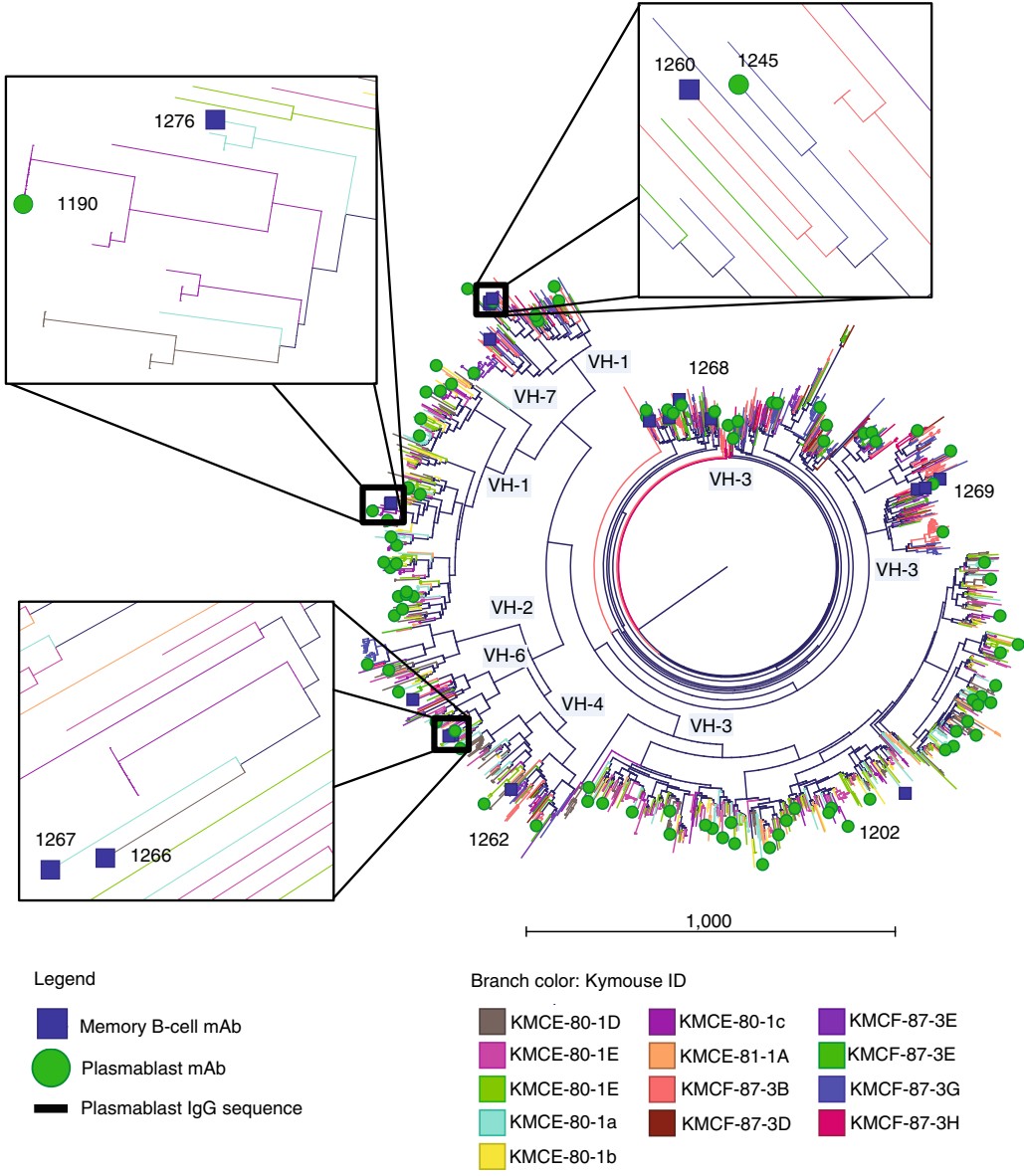

**Fig. 1** Phylogram representation of B-cell responses to Pfs25-VLP immunization. Natively paired H- and L-chain, full-length, variable region sequences from 13 Pfs25-VLP-immunized Kymice were combined into one phylogram based on full-variable region H- and L-chain amino acid sequence similarity among plasmablast IgG and memory B-cell IgG sequences. The 95 plasmablast-derived and 18 memory B-cell-derived sequences selected for recombinant antibody expression are noted as green circles and blue boxes, respectively. Recombinant antibodies listed in Table 1 chosen for further analysis are labeled. Between pairs of nodes, sum of radial distances is proportional to the Kimura distance as indicated by the scale bar. Branch color denotes the specific animal from which the sequence was derived. Higher magnification insets show cases in which two different anti-Pfs25 antibodies of highly similar sequence originate from different individual Kymice and, in two cases, different antibody-expressing cell types. The subset of plasmablast sequences from Pfs25-VLP immunized Kymice that are similar to sequences from empty VLP-immunized Kymice are included as part of the 1564 plasmablast data set shown in the tree, but were not used as sources for recombinant antibody expression

**Table 1 Variable gene usage of 10 SMFA-active Pfs25-specific memory B cells (MBC)-derived and splenic plasmablasts (PB)-derived mAbs**

| mAb | VH | DH | JH | HCDR3 length | SHM nt | SMH aa | VK/L | JK/L | LCDR3 length | SHM nt | SHM aa | Source | Mouse |
|-----|------|------|----|--------------|--------|--------|------|------|--------------|--------|--------|--------|---------|
| 1190 | 1–2 | 2–2 | 4 | 13 | 13 | 9 | 3–21 | 3 | 12 | 9 | 6 | PB | E-80-1c |
| 1276 | 1–2 | 2–2 | 4 | 13 | 9 | 7 | 3–21 | 3 | 10 | 12 | 5 | MBC | E-80-1a |
| 1245 | 1–18 | 4–17 | 6 | 18 | 14 | 9 | 2–30 | 4 | 9 | 5 | 2 | PB | F-87-3G |
| 1260 | 1–18 | n.d | 6 | 18 | 17 | 9 | 2–30 | 4 | 9 | 2 | 1 | MBC | F-87-3b |
| 1266 | 6–1 | 6–19 | 4 | 16 | 8 | 2 | 3–10 | 2/3 | 11 | 4 | 3 | MBC | E-80-1d |
| 1267 | 6–1 | 6–19 | 4 | 16 | 7 | 4 | 3–10 | 3 | 11 | 6 | 3 | MBC | E-80-1a |
| 1202 | 3–11 | 3–10 | 3 | 15 | 15 | 8 | 3–1 | 1 | 9 | 10 | 5 | PB | E-80-1c |
| 1262 | 4–4 | 4–17 | 4 | 13 | 15 | 11 | 1–9 | 2 | 9 | 5 | 3 | MBC | F-87-3b |
| 1268 | 3–21 | 3–10 | 5 | 17 | 5 | 4 | 1–9 | 1 | 9 | 3 | 3 | MBC | F-87-3G |
| 1269 | 3–23 | 3–10 | 6 | 18 | 11 | 7 | 3–20 | 5 | 6 | 4 | 3 | MBC | F-87-3F |

sequence diversity between isolates is thought to be the result of limited immune selective pressure[14,15]. Antibody targeting of Pfs25 can result in a significant reduction in the number of oocysts in in vitro membrane feeding assays[16,17]. The high sequence conservation within *Pf* strains and the fact that antibodies taken up by mosquitoes in blood meals can impede parasite development make Pfs25 an attractive TBV target. As a result, Pfs25 has been a leading target for vaccine design[18,19].

The main challenge associated with a TBV is to elicit by human vaccination, sufficient titers of potent antibodies to inhibit the parasite in the mosquito gut after a blood meal[20]. Past attempts using Pfs25 as an immunogen have suffered from low immunogenicity despite the use of various adjuvants designed to boost the humoral antibody response[6,21]. A high-resolution definition of sites of vulnerability on Pfs25 would enable the structure-guided design of immunogens that might increase the immunogenicity of potent epitopes.

Here we characterize monoclonal antibodies (mAbs) elicited from the Kymouse human immunoglobulin (Ig) loci transgenic mice, immunized with recombinant plant-produced Pfs25 virus-like particles (VLPs)[22–24]. We delineate the atomic structure of Pfs25 as recognized by six mAbs covering two distinct regions defining functionally important epitopes. Using this information, we identify the antibodies most effective at inhibiting oocyst development and show that two non-overlapping epitope regions can be targeted additively to lower individual antibody titers required for parasite inhibition.

## Results

**Pfs25-specific mAbs derived from humanized mice.** To develop a molecular understanding of the antibody response to Pfs25, Kymab mice (Kymouse^TM), that are transgenic for the non-rearranged human antibody germline repertoire, were immunized with a plant-produced Pfs25-VLP immunogen[23] (Supplementary Figs. 1 and 2). MAbs against Pfs25 were generated using direct sequencing of expressed IgG mRNA from both Pfs25-specific memory B cells following antigen-specific single-cell sorting and from splenic plasmablasts without any pre-selection (Supplementary Fig. 3). The anti-Pfs25 response in both cell types was assessed to determine which compartment contains the most functionally potent, active, or broad set of antibodies. Five hundred fifty-five memory B-cell-derived mAbs were generated as human IgG1 and screened for binding to Pfs25 by homogeneous time-resolved fluorescence and surface plasmon resonance. Two hundred twenty-five mAbs were confirmed by both assays to bind Pfs25, and had affinities ranging from ~500 nM to less than 1 nM (Supplementary Fig. 3B). In parallel, 1564 natively paired, complete variable region sequences were derived from plasmablasts. All plasmablast sequences were combined with the memory B-cell sequences available to allow a comprehensive gene usage analysis

of both B-cell repertoires from 13 Kymice in response to immunization with Pfs25-VLP (Fig. 1). This analysis revealed a diverse response, spanning 784 Ig lineages across 6 *VH* gene families (Fig. 1). Interestingly, anti-Pfs25 antibodies of highly similar sequence were observed in different individual Kymice, with 102 Ig common lineages being present in two or more similarly immunized Kymice (Supplementary Fig. 3D). As expected, common Ig lineages were also observed in different antibody-expressing cell types (memory B cells and plasmablasts) (Fig. 1 and Supplementary Fig. 3C).

Ninety-five plasmablast-derived mAbs, representing 93 unique, putative lineages, and 18 memory B-cell-derived mAbs were generated after downselection using sequence analyses and Ig lineage clustering as criteria (described in "Methods" section). Thirty-four of the 95 plasmablast-derived mAbs were confirmed to bind to Pfs25-VLP and not empty VLP by ELISA (Supplementary Fig. 3C). Only seven plasmablast-derived mAbs bound to soluble Pfs25 indicating most required avidity for binding, or Pfs25 on VLPs has epitopes not present in soluble Pfs25 (Supplementary Fig. 3C). Based on their binding reactivity to Pfs25-VLP and/or soluble Pfs25, 35 mAb derived from 20 plasmablast and 15 memory B cell mAbs were selected for further functional activity in the membrane feeding assay. At a single high concentration of 375 μg/ml (or lower, as one antibody had limiting material), 19 showed inhibition in a single concentration functional screen above 80% and one greater than 50% (Supplementary Table 1). These 20 mAbs were derived from 15 independent lineages with three only found in plasmablasts, six derived from lineages in both plasmablasts and memory B cells, and six only from memory B cells. Ten of these antibodies were selected for further analysis (Table 1). Based on their germline gene usage, apparent recombination and somatic hypermutation, the 10 antibodies bin into seven unique groups (Fig. 1 and Table 1). Three pairs of antibodies (1190/1276, 1245/1260, and 1266/1267) constitute examples of common selection of sequence features in the immune response of similarly immunized Kymice. Within this subset, two mAb pairs (1190/1276 and 1245/1260) derive from memory B cells and plasmablasts pools, providing examples of similar B cell diversity being present in both immune repertoires for SMFA-active mAbs (Table 1).

**Binding competition reveals two antibody recognition sites.** To determine the putative-binding sites of the 10 anti-Pfs25 mAbs, we expressed antigen-binding fragments (Fabs) of the antibodies and performed real-time competition-binding studies by biolayer interferometry (Fig. 2). The tested antibodies were then binned according to their competition groups. 4B7 IgG previously had been mapped to the B loop of the EGF-like domain 3[25] and was included in the epitope-binning studies as a comparison. Epitope-binning studies identified two distinct antibody recognition sites

| | | | Site 1a | | | Site 1b | Site 1c | | | Site 1d | Site 2 | |
|---|---|---|---|---|---|---|---|---|---|---|---|---|
| | | 1190 | 1276 | 1262 | 4B7 | 1269 | 1202 | 1266 | 1267 | 1268 | 1245^ | 1260^ |
| Site 1a | 1190 | 0 | 0 | 0 | 15 | 0 | 104 | 84 | 84 | 80 | 42 | 34 |
| | 1276 | 0 | 0 | 5 | 26 | 0 | 91 | 72 | 78 | 73 | 27 | 18 |
| | 1262 | 0 | 0 | 0 | 8 | 0 | 82 | 83 | 82 | 70 | 68 | 83 |
| | 4B7 | 28 | 27 | 27 | 7 | 34 | 28 | 35 | 44 | 121 | 78 | 86 |
| Site 1b | 1269 | 0 | 0 | 0 | 6 | 0 | 0 | 19 | 61 | 77 | 54 | 41 |
| | 1202* | 72 | 66 | 54 | 50 | 33 | 18 | 27 | 34 | 58 | 52 | 9 |
| Site 1c | 1266* | 65 | 59 | 65 | 49 | 41 | 0 | 7 | 16 | 17 | 67 | 34 |
| | 1267 | 90 | 89 | 87 | 46 | 94 | 0 | 0 | 0 | 0 | 87 | 63 |
| Site 1d | 1268* | 65 | 62 | 49 | 65 | 78 | 71 | 0 | 0 | 0 | 48 | 5 |
| Site 2 | 1245 | 101 | 103 | 103 | 94 | 109 | 131 | 134 | 131 | 135 | 0 | 0 |
| | 1260 | 83 | 69 | 82 | 87 | 80 | 94 | 92 | 88 | 82 | 8 | 0 |

**Fig. 2** Epitope binning of anti-Pfs25 Kymouse-derived human antibodies. Primary antibodies tested are listed in the left column, while secondary competing antibodies are listed at the top in rows. Data indicate the percent of competing antibody binding compared to the maximum competing antibody response in the absence of the primary antibody. Boxes are colored according to competition status. Competing antibodies that displayed ≤ 33% maximal binding are colored black and considered competing, between 34 and 66% binding are considered intermediate and colored gray, and those that displayed ≥ 66% binding are colored white and considered non-competing. Epitope bins are listed above or beside the table, with site 1 competing antibodies colored in shades of blue and site 2 competing antibodies colored green. *Primary antibodies displayed a fast off-rate, resulting in intermediate binding values for true non-competing antibodies. ^Competing antibodies displayed a slow on-rate, leading to low values. The well-characterized mouse mAb 4B7 is highlighted in yellow as a reference

on Pfs25, that we term sites 1 and 2. Site 1 is complex and is recognized by eight of the 10 antibodies with four overlapping sites, labeled a to d, highlighting a fine specificity for antigen recognition of this immunogenic Pfs25 region. Six of the eight site 1 antibodies were in competition with 4B7 (site 1a and b antibodies), likely indicating that site 1 antibodies bound to the EGF-like domain 3 (Fig. 2). In contrast, the site 2 bin is completely distinct from site 1; it contains two antibodies, 1245 and 1260, and showed no interference with any of the other tested antibodies. Importantly, antibodies with similar gene usage (1190/1276, 1245/1260, and 1266/1267) segregated into the same bins, providing further evidence of correct binning per epitope results. Thus, SMFA-active antibodies recovered after immunizing Kymice with Pfs25-VLP immunogens bound two distinct binding sites, one likely centered around the EGF-like domain 3 and one new as yet uncharacterized site.

**Binding kinetics characterization of elicited antibodies**. To understand the binding characteristics of elicited anti-Pfs25 SMFA-active antibodies, we performed binding kinetics analyses of the 10 Fabs to Pfs25 by biolayer interferometry (Supplementary Fig. 4). Fabs displayed an 18-fold difference in affinities for Pfs25, ranging from 3.7 to 67 nM (Supplementary Fig. 4K and Supplementary Table 2). This spread is reflected by association rates ranging from $2.2 \times 10^4$ to $4.8 \times 10^5$ $1\,M^{-1}\,s^{-1}$

(Supplementary Fig. 4L and Supplementary Table 2), and dissociation rates ranging from $1.6 \times 10^{-2}$ to $3.8 \times 10^{-4}\,s^{-1}$ (Supplementary Fig. 4M and Supplementary Table 2). Antibodies 1269 ($K_D = 3.7$ nM) and 1245 ($K_D = 31$ nM) bound site 1 and site 2, respectively, with the highest affinity within their epitope bins. Antibody recognition of site 2 was associated with slower on-rates and slower off-rates. The Fab affinities observed here corroborate the binding that led to the identification of these antibodies in initial screening (Supplementary Fig. 3B). We also note that as IgG's, 1269 and 1245 bind with similar avidity to Pfs25 as the 4B7 IgG (Supplementary Fig. 5). Their extremely slow off-rates prevent accurate determination of their binding avidities by biolayer interferometry, which we estimate to be $<10^{-12}$ M.

**Crystal structures of Pfs25 in complex with six mAbs**. To understand the fine specificity of Pfs25 antibody recognition against sites 1 and 2, we attempted to determine co-complex crystal structures of all antibodies tested in competition binding studies. All antibodies recognized conformational epitopes, and unlike mAb 4B7 did not react with linear peptides of the Pfs25 protein, or with denatured Pfs25 in western blot experiments. As such, we focused our structural efforts on antibody–Pfs25 complexes. We obtained the structures of four site 1a antibodies, 1190, 1262, 1269, and 1276, in complex with Pfs25 to 2.4, 2.7, 2.5, and 2.2 Å resolution, respectively; and two site 2 antibodies, 1245

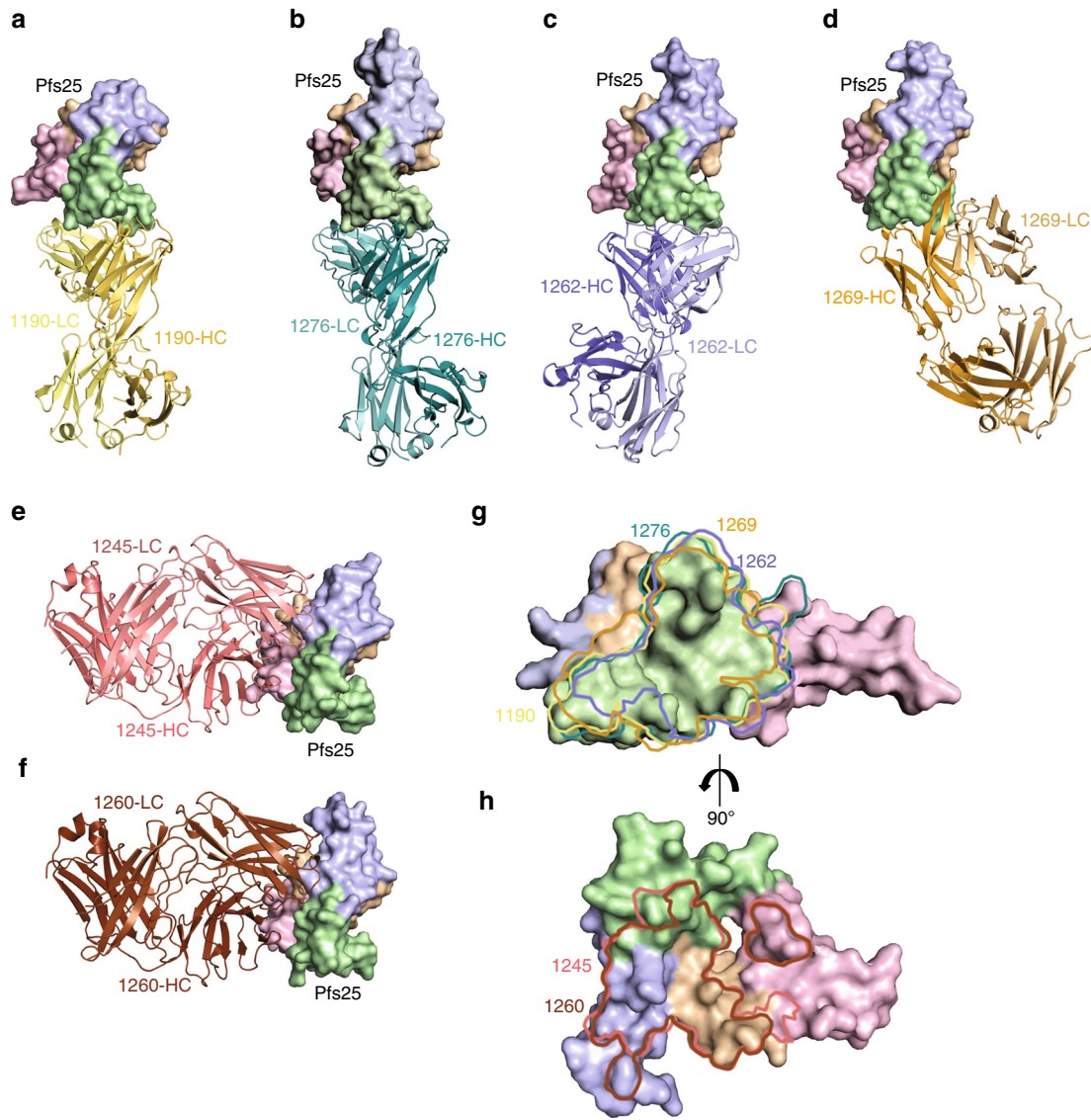

**Fig. 3** Crystal structures of six mAbs in complex with Pfs25. Crystal structures of **a** 1190, **b** 1276, **c** 1262, **d** 1269, **e** 1245, and **f** 1260 in complex with Pfs25. All crystal structures are shown according to the same Pfs25 orientation. Pfs25 is represented as surface and EGF-like domains 1–4 are colored in wheat, pink, green, and blue, respectively. **g** Site 1a epitopes for 1190, 1262, 1269, and 1276 antibodies traced onto the surface of Pfs25. **h** Site 2 epitopes for 1245 and 1260 antibodies traced onto the surface of Pfs25

and 1260, in complex with Pfs25 to 1.9 and 3.3 Å resolution, respectively (Fig. 3a–f and Supplementary Tables 3–9). These crystal structures provide a broad molecular understanding of B cell responses elicited by the Pfs25-VLP immunogen in Kymice. As indicated in the competition-binding data, site 1a antibodies bound to the EGF-like domain 3 (Fig. 3g, green domain), while site 2 antibodies bound to the triangular face of Pfs25—a previously uncharacterized antigenic site of Pfs25—contacting all four EGF-like domains (Fig. 3h).

These crystal structures represent the first three-dimensional structural delineation of the TBV Pfs25 antigen. Comparison of the structures of Pfs25 in each complex to the previously determined Pvs25 structure (Supplementary Fig. 6A)[13] showed that Pfs25 adopts a similar overall structure as Pvs25, in which four distinct EGF-like domains assume a triangular arrangement and disulfide-bonding patterns are conserved (Supplementary Fig. 6B). The six Pfs25 structures superposed well, displaying a low average root mean square deviation (RMSD) of 0.8 Å. The RMSD was highest in the loop regions of the EGF-like domains

(Supplementary Figs. 6C and 7). Differences between the six Pfs25 structures and Pvs25 (Supplementary Fig. 6D) were considerably greater (overall RMSD = 2.1 Å), particularly in the EGF-like domain 4 (RMSD = 4.0 Å, Supplementary Fig. 6C). Of note, residues that promote the characteristic triangular architecture are either conserved between Pfs25 and Pvs25, or in the case of residues Glu/Lys22 and Lys/Glu98, are swapped between homologs therefore adopting similar conformations (Supplementary Fig. 6E).

**Specificities of site 1-directed mAbs**. Site 1a mAbs 1190, 1262, 1269, and 1276 recognize the EGF-like domain 3 of Pfs25 and compete for largely overlapping recognition sites (Fig. 4a). We compared the angle of approach of site 1a mAbs by calculating and comparing the center of mass of the variable domains of the antibodies with their respective Pfs25-interacting residues (Fig. 4b), and noted varied binding modes for recognition of the EGF-like domain 3. MAbs 1190 (plasmablast-derived from

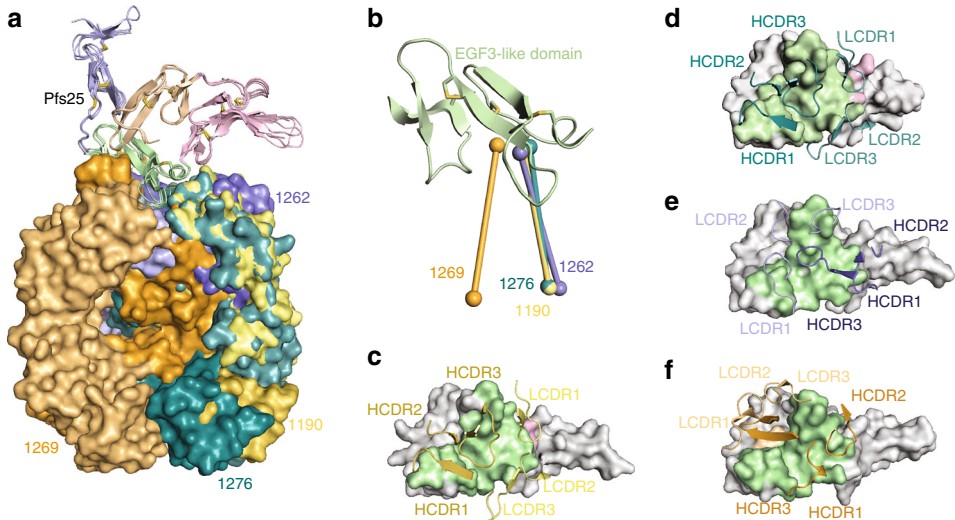

**Fig. 4** Molecular details of site 1a antibody recognition. **a** Superposition of site 1a antibody–Pfs25 co-complex crystal structures. Pfs25's are shown as cartoon and colored according to EGF-like domain as in Fig. 3. 1190, 1262, 1269, and 1276 Fabs are shown as surface representation and colored in yellow, blue, orange, and teal, respectively. **b** Comparison of the angle of approach. The center of mass for both the Fab variable domains and interacting Pfs25 residues are shown as spheres and colored as in **a**. CDR loops that interact with Pfs25 for **c** 1190, **d** 1276, **e** 1262, and **f** 1269. Non-interacting Pfs25 residues are colored gray, while interacting residues are colored according to their EGF-like domain as in **a**

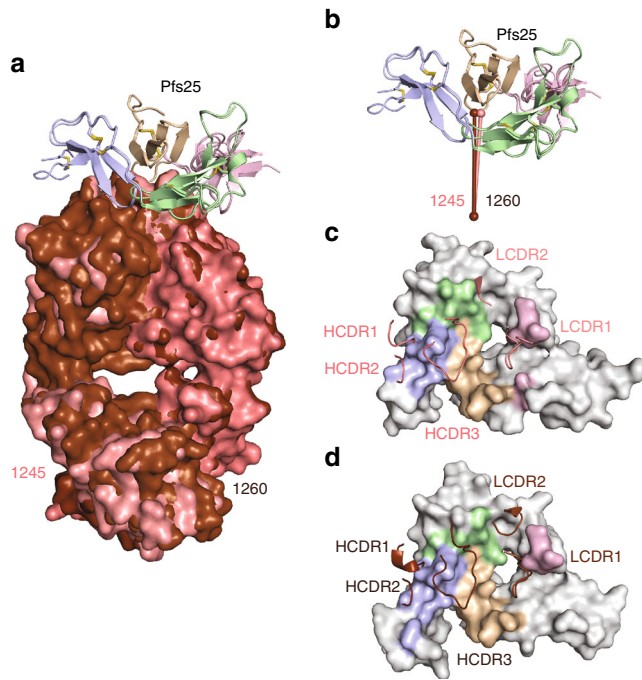

**Fig. 5** Molecular details of site 2 antibody recognition. **a** Superposition of site 2 antibody–Pfs25 co-complex crystal structures. Pfs25s are shown as cartoon and colored according to EGF-like domain as in Fig. 3. 1245 and 1260 Fabs are shown as surface representation and colored in salmon and brown, respectively. **b** Comparison of the angle of approach. The center of mass for both the Fab variable domains and interacting Pfs25 residues are shown as spheres and colored as in **a**. CDR loops that interact with Pfs25 for **c** 1245 and **d** 1260. Non-interacting Pfs25 residues are colored gray, while interacting residues are colored according to their EGF-like domain

Kymouse E-80-1c) and 1276 (memory B cell-derived from Kymouse E-80-1a) are part of a mAb pair, and differ by eight H-chain and 14 L-chain residues. Of these residues, two in the HCDR2 (Thr/Asn53 and Asp/Ser54) and one each in the LCDR2 (Tyr/Phe52) and LCDR3 (His95B/Arg95) contact Pfs25.

Accordingly, 1190 and 1276 demonstrated similar interacting buried surface areas (BSA, 1136.8 and 1104.0 Å$^2$) and hydrogen (H)-bonding networks (17 and 14 H-bonds, respectively) (Fig. 4c, d and Supplementary Tables 4 and 5). In addition, their H- and L-chains contributed almost equally to Pfs25 contacts and BSA (Supplementary Table 10). The slightly greater BSA and H-bonds of 1190 compared to 1276, likely explain the higher affinity 1190 exhibits compared to 1276 (6.7 nM vs. 16 nM, Supplementary Table 2). Thus, 1190 and 1276 bind to Pfs25 similarly, and approach Pfs25 from the same direction, contacting predominantly EGF-like domain 3 residues, but also residues 75, 76, 77, and 78 from the EGF-like domain 2 (Supplementary Tables 4 and 5). 1190 and 1276 are examples of paratope fit driving convergent selection in the germinal centers of two different Kymice (Table 1).

Despite a rotation of ~173° compared to 1190/1276, mAb 1262 only shifts its center of mass by 3.5° effectively swapping its variable L- and H-chain domains with the H- and L-chains of 1190/1276 (Fig. 4b). 1262 contacts EGF-like domain 3 residues exclusively (Fig. 4e, Supplementary Table 6). 1262 exhibited the lowest BSA of the site 1a mAbs (929.6 Å$^2$), with the H-chain contributing slightly more BSA (534.8 Å$^2$ vs. 394.8 Å$^2$) and H-bonds (9 vs. 6) compared with its L-chain (Supplementary Table 10). In contrast, mAb 1269 demonstrates an angular difference of 31° and 35° in its angle of approach compared to 1190/1276 and 1262, respectively, resulting in 1269 contacting EGF-like domain 1 residues 18 and 19 (Fig. 4b, f and Supplementary Table 7). In addition, contacts between antibody 1269 and Pfs25 are dominated by its H-chain, which contributes 889.5 Å$^2$ of BSA and 17 H-bonds compared to its L-chain (239.5 Å$^2$ and two H-bonds). As 1269 is the only site 1a mAb that co-resides in the site 1b and 1c bins, it is likely that site 1b and 1c mAbs also recognize these unique EGF-like domain 1 residues of Pfs25. MAb 1269 exhibited the greatest number of H-bonds of the site 1a mAbs, which together with its high BSA, helps explain its highest-binding affinity for Pfs25 compared to the other site 1a mAbs (Supplementary Fig. 4 and Supplementary Tables 2 and 10). Interestingly, an intricate H-bonding network between the HCDR2 of 1269 and Pfs25 residue Glu88 led to a unique rotameric state for this antigen residue compared to recognition

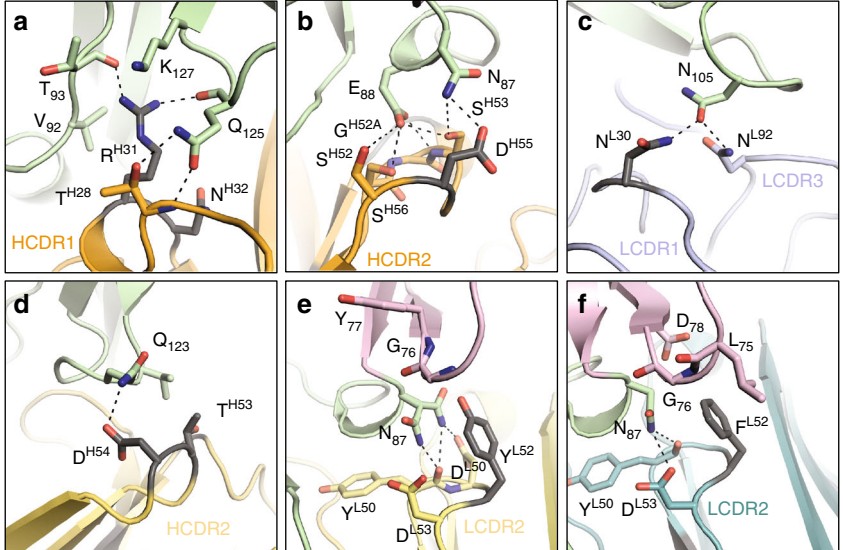

**Fig. 6** Affinity maturation of mAbs at the binding interface. Somatic hypermutations (SHMs) that introduce new contacts with Pfs25 are colored dark gray. **a** A S31R mutation in the HCDR1 of 1269 introduces two H-bonds with Pfs25 residues T93 and Q125. **b** A G55D mutation in the HCDR2 of 1269 introduces an additional H-bond with Pfs25 residue N87. **c** A S30N mutation in the LCDR1 of 1262 leads to an additional H-bond with Pfs25 residue N105. **d** A S54D mutation in the HCDR2 of 1190 leads to an additional H-bond contact with Pfs25 residue Q123. **e** A S52Y mutation in the LCDR2 of 1190 leads to additional van der Waals interactions with Pfs25 residues, and ensures that N87 hydrogen bonds with adjacent LCDR2 residues. **f** Similar to 1190, 1276 has a S52F mutation in the LCDR2 that leads to additional van der Waals contacts with Pfs25 residues, and ensures that N87 hydrogen bonds with adjacent LCDR2 residues

by the other three antibodies against the same sub-site, resulting in a more electronegative surface targeted by 1269 (Supplementary Fig. 8). Crystals for Pfs25 in complex with site 1b, 1c, and 1d mAbs 1202, 1266, 1267, and 1268 remain elusive, and therefore the exact composition of their epitopes is still to be uncovered. In summary, the site 1a antibodies described here contact 17 common residues, which represent a critical immunogenic site on Pfs25 (Supplementary Fig. 9).

**Specificities of site 2-directed mAbs.** Site 2 mAbs 1245 (plasmablast-derived from Kymouse F-87-3G) and 1260 (memory B cell-derived from Kymouse F-87-3b) are examples of convergent selection in two different Kymice (Table 1), and share 84 and 96% sequence identity in the variable regions of their H- and L-chains, respectively. Interestingly, our crystal structures reveal that none of the differing amino acids contact Pfs25 with their sidechains. Therefore, 1245 and 1260 bind to Pfs25 very similarly, adopt a similar angle of approach to one another, and importantly, contact all four EGF-like domains (Fig. 5a, b). 1245 and 1260 demonstrated a smaller interaction site compared to site 1a mAbs, with BSAs of 803.1 Å² and 847.6 Å², respectively (Supplementary Table 10). The smaller interface likely contributed to their lower binding affinities (Supplementary Fig. 4 and Supplementary Table 2). Their interactions with Pfs25 were predominantly mediated by their HCDR3 (Fig. 5c, d and Supplementary Tables 8 and 9), which contributed eight out of the total 14 H-bond partners for 1245, and seven out of the total 12 H-bond partners for 1260 (Supplementary Table 10). Notably, the LCDR3 of these antibodies did not contact Pfs25 (Fig. 5c, d). We also note that none of the site 1a and site 2 mAbs described here bind to the antigenic site recognized by the Pvs25-reactive 2A8 mAb (Supplementary Fig. 10).

**Role of somatic hypermutations in anti-Pfs25 mAbs.** To determine the role of affinity maturation in Pfs25 recognition, we analyzed the sequence differences between the mature antibodies and their germline precursors, which is known for the Kymouse.

Our analysis reveals that mAbs underwent between 2 and 11 somatic hypermutations (SHM) in their H-chains and between one and six SHMs in their L-chains compared to germline (Table 1). Analysis of the site 1a and site 2 mAb crystal structures demonstrates the effect these mutations have on Pfs25 recognition. Site 1a mAbs typically acquired 1–2 mutations that led to additional H-bonds or van der Waals interactions (Fig. 6). In mAb 1269, this includes a S31R mutation in the HCDR1 and a G55D mutation in the HCDR2 that introduce three and one H-bonds, respectively (Fig. 6a, b). In mAb 1262, a S30N mutation in the LCDR1 leads to an additional H-bond with Pfs25 residue Asn105 (Fig. 6c). In mAb 1190, a S54D mutation in the HCDR2 leads to an additional H-bond (Fig. 6d), while both 1190 and 1276 each have an S52F/Y mutation that introduces van der Waals contacts with multiple EGF-like domain 2 residues, and ensures that Asn87 adopts a favorable conformation to H-bond with other LCDR2 residues (Fig. 6e, f). In contrast and somewhat surprisingly, site 2 mAbs 1245 and 1260 apparently displayed no SHM that led to additional contacts with Pfs25. However, changes in framework regions may have contributed to reshaping the antibody paratopes leading to increased affinity. Additionally, both 1245 and 1260 interact with Pfs25 predominantly through their HCDR3; therefore, our inferred analysis of SHM may have missed important B cell ontogeny mutations in the 5–6 residues encoded in the non-templated VD and DJ junctions within the HCDR3.

**mAbs against the two sub-sites are additive in SMFA.** We next quantified the inhibition potency of the mAbs, and evaluated whether combinations of the mAbs that target sites 1 and 2 would exhibit additive or synergistic inhibition activity in SMFA. We determined the specific potency of the two mAbs with the highest binding affinity against sites 1 and 2, 1269 and 1245, respectively (Feed #1 and #2; Fig. 7a and Supplementary Table 11). In agreement with previous studies with monoclonal and polyclonal antibodies[26–28], there was a linear dose response for each mAb when the ratio of mean oocyst count (right *y*-axis, on a log-scale)

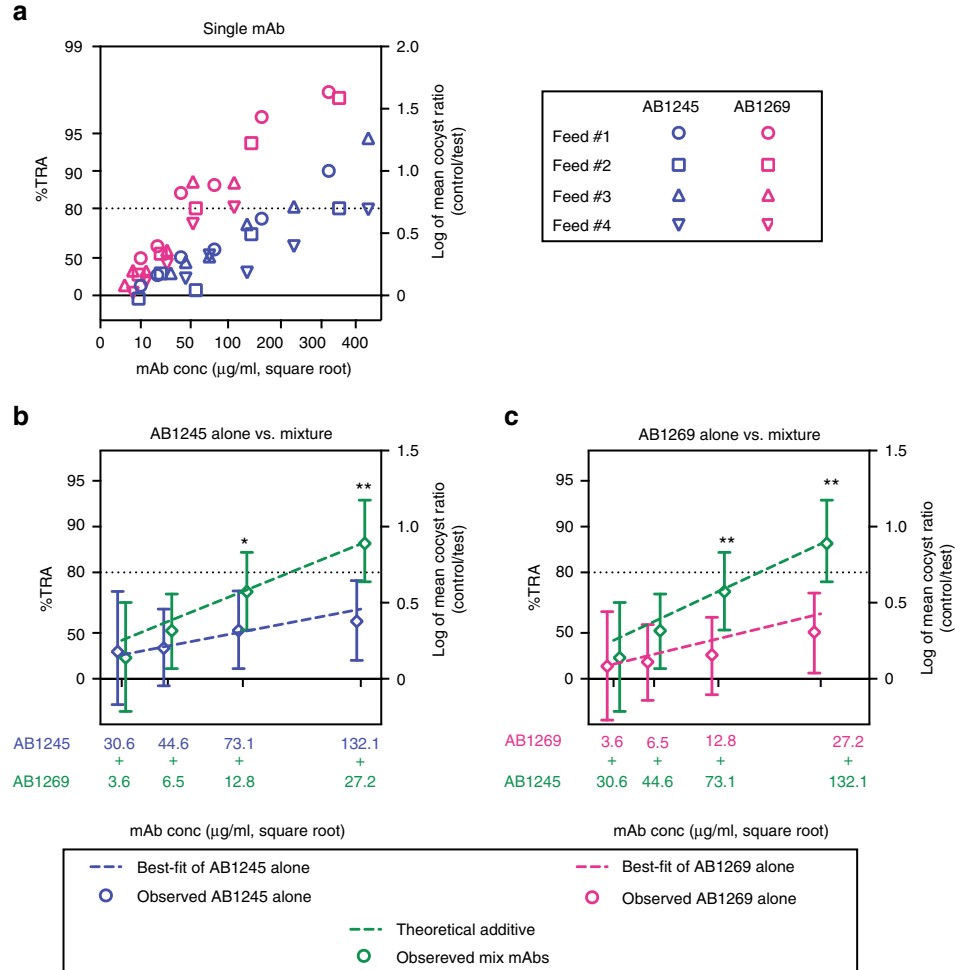

**Fig. 7** MAbs 1245 and 1269 display an additive effect in SMFA. **a** 1245 and 1269 antibodies were tested at multiple concentrations in four independent assays. Each antibody concentration is shown on a square root scale (*x*-axis). The associated % inhibition (%TRA) value is shown on the left side of the *y*-axis. The ratio of mean oocyst (mean oocyst density in control divided by mean in test) is plotted on a log scale (log mean ration, right side of *y*-axis). **b, c** 1245 and 1269 antibodies were tested at indicated concentrations either alone or a mixture of the two mAbs. The best-estimated %TRA and the 95% CIs of mean oocyst density calculated from two independent assays are shown (observed). In addition, the best-fit line of each mAb and the theoretical additive values (assuming the two antibodies work additively) are shown as dotted lines. **b** is depicted based on the 1245 antibody concentration (*x*-axis) and **c** by 1269 antibody concentration, and the same SMFA data from mixture groups are plotted in both figures. An asterisk indicates that the mixture group showed significantly higher %TRA than the corresponding single mAb group (*<0.05; **<0.01). The *p* values were calculated using the zero-inflated negative binomial model described previously[50]

was plotted against the square root of antibody concentration (*x*-axis). MAb 1269, with an IC$_{80}$ of 63 μg/ml (95% CI, 53–75), was significantly more potent than 1245, with an IC$_{80}$ of 263 μg/ml (95%CI, 211–348), in accord with the higher binding affinity of 1269 (Supplementary Table 2). IC$_{80}$ was used as an adequate measure because the SMFA assay becomes highly variable below the 80% inhibition level. In comparison, the IC$_{80}$ of 4B7 was 29 μg/ml (95% CI, 24–34).

To design combination experiments for mAbs against sites 1 and 2, we determined concentrations at which both mAbs show similar levels of transmission-reducing activity (%TRA). Since 1269 was more potent than 1245 (Fig. 7a), higher concentrations of 1245 than 1269 were used in subsequent co-administration feeds. At the lowest concentration, a mixture of 31 μg/ml of 1245 and 4 μg/ml of 1269 were fed to mosquitoes (31-4 mixture group). These individual antibody concentrations correspond to 37% TRA when tested alone. Following the same scheme, three more mixture groups were tested: 45-7 (45% TRA by either single antibody), 73-13 (56% TRA), and 132-27 (69% TRA) mixture groups. In the next feeding assays, the antibodies were fed either

alone or as mixtures (Feed #3 and #4; Fig. 7b, c and Supplementary Table 12). At the higher doses (73-13 and 132-27 combinations), the two mAbs showed significantly higher %TRA than the corresponding single antibody groups (up ~20–30%, Fig. 7b, c). Since the SMFA has lower precision for weaker activity samples[26,28], the results of co-administration at the lower doses (31-4 and 45-7 combinations) are less certain, but still trend toward higher %TRA when the two mAbs are co-administered. In summary, we interpret our results to indicate that the two mAbs worked additively, as the observed %TRAs in the mixture groups were clearly similar to the levels expected for a theoretical additive effect (Fig. 7b, c).

## Discussion

An effective TBV for malaria will need to elicit high titers of potent antibodies in humans to effectively inhibit parasite development in mosquitoes[6]. Our molecular and functional analyses of the antibody response from Kymice Ig-humanized mice after immunization with plant-expressed Pfs25-VLP reveal

two distinct immunogenic sites on Pfs25. Site 1 broadly overlaps the well-characterized 4B7 epitope in the EGF-like domain 3, while site 2 is a newly identified immunogenic site that spans the four EGF-like domains of Pfs25. Our highest-affinity SMFA-active mAb against site 1, 1269 (3.7 nM affinity as Fab, $<10^{-12}$ M avidity as IgG), is more potent in TRA than the highest-affinity SMFA-active mAb isolated against site 2, 1245 (31.0 nM affinity as Fab, $<10^{-12}$ M avidity as IgG).

Our molecular characterization of functional mAbs against Pfs25 informs on the arrangement of the antigen on the surface of ookinetes. It is reasonable to assume that antibody inhibition of the development of ookinetes can only occur if their epitopes on Pfs25 are exposed and accessible on the cell surface. Saxena et al.[13] had previously suggested a model that featured Pvs25 coating the cell surface of ookinetes in a tile-like fashion inferred from crystal packing interactions combined with residue conservation analyses. Our structural data shows that site 1-directed mAbs (e.g., 1269), which are active in SMFA, recognize a face of Ps25 that is buried and inaccessible at the interface of the Pvs25 tile-like model (Supplementary Fig. 11), arguing against the tile-like model for Pfs25. The epitopes of site 2-directed mAbs (e.g., 1245) are somewhat accessible in the tile-like model (Supplementary Fig. 11), but the fact that site 2 mAbs are less potent inhibitors than site 1 mAbs further argues against this arrangement of Pfs25 on the ookinetes surface.

The Kymouse provides a model system for the characterization of B cell responses in a humanized immune setting and have been successful in this regard in diseases such as HIV[24]. Here we report to our knowledge the first crystal structures of Kymouse-derived antibodies to their cognate antigen. Our results support the wide diversity of the B cell repertoire reported for the Kymouse[22]. Memory B cell- and plasmablast-derived mAbs displayed a varied gene usage, bound to at least two distinct immunogenic sites, displayed binding affinities spanning the nanomolar range, and underwent affinity maturation, indicating $T_{FH}$ cell help in germinal centers[29]. The mAbs also showed a range of BSAs to both predominantly single-domain (antigenic site 1a) and more elaborate multi-domain (antigenic site 2) epitopes, as well as recognition modes that employ equal H- and L-chain contributions (1190 and 1276), or H-chain-dominated interactions (1245, 1260, and 1269). Our data show that both memory B cell- and plasmablast-derived mAbs contain functional antibodies. It is therefore likely they both contribute to the serum polyclonal response (from short- and long-term plasma cells) and therefore may also be part of a long-lived response, which is an important goal in the development of a durable TBV. Characterization of the antibody response to the same Pfs25 immunogen in humans will be an important next step to compare and further validate the Kymouse as a predictive model system to evaluate immunogenicity of TBV candidates.

Our six crystal structures revealed that the loop regions of the EGF-like domains, including the 4B7 loop, are not rigid; instead, they are capable of adopting multiple conformations, providing important knowledge for immunogen design seeking to elicit B cell responses against the most potent Pfs25 conformation. Importantly, both Pfs25 immunogenic sites uncovered in our studies can be targeted simultaneously and when 1269 and 1245 are co-administered in SMFA, they act additively to reduce the concentration of individual antibody required to achieve the same transmission-reducing activity. We note, however, that 1269 is considerably more potent than 1245, therefore the required concentration for TBV activity for 1269 alone is much lower than for a 1245/1269 mixture. It is possible that site 2 mAbs of equal potency to site 1 mAbs exist but were not identified in our studies and such antibodies might be revealed by additional data mining. Additionally, it is possible that other sites exist that also elicit

potent mAbs. Indeed, it has been shown that EGF-like domain 2 is sensitive to the transmission-blocking activity of antibodies, even at low titers compared to other antigenic sites on Pfs25[8]. Nevertheless, our data currently support immunogen design approaches that focus the immune response on antigenic site 1 (e.g., ref. [30]) to preferentially elicit 4B7- and 1269-like mAbs, while also highlighting the benefits of polyclonal responses that can act additively. An understanding of the Pfs25 three-dimensional structure and the location of protective epitopes will also enable the optimal display and density of Pfs25 on nanoparticles, which is known to significantly boost B cell responses[31–34].

Analogous to other disease models, such as respiratory syncytial virus (RSV)[35,36] and HIV[31,37,38], the crystal structures presented here can serve as blueprints for epitope-focusing efforts and provides critical insights to guide the design of malaria transmission-blocking therapeutic or prophylactic antibodies.

## Methods

**Antigens and immunizations.** Antigen was provided as a VLP with surface-displayed Pfs25 (Fraunhofer CMB, Newark, DE). An "empty" VLP preparation was also supplied to act as a control for any non-Pfs25 antigens present on the VLP vehicle. Monomeric Pfs25 was generated in *Nicotiana benthamiana* as previously described[39]. Fourteen HK (five females and nine males) and fourteen HL (five females and nine males) 9-week-old mice were immunized[22]. All mice were maintained, and all procedures carried out under United Kingdom Home Office License 70/8718 and with the approval of the Sanger Institute Animal Welfare and Ethical Review Body. The immunization regimen consisted of three IP immunizations performed on days 0, 21, and 42 using 1 µg of Pfs25-VLP formulated in Montanide ISA720 (70% v/v, Seppic Inc), and a final boost at day 72 with 1 µg of unadjuvanted Pfs25-VLP, also administered IP. Terminal peripheral blood and splenocytes were collected on day 79. Polyclonal antibody responses directed to Pfs25-VLP, empty VLP, and soluble Pfs25 were determined throughout the immunization phase by ELISA. Upon completion of the immunization regimen, all mice had developed detectable anti-Pfs25 antibody titers. Transmission-reducing activity (TRA) was determined by SMFA using polyclonal mouse sera and monoclonal antibodies that bound Pfs25.

**Identification and selection of Pfs25-specific mAb sequences.** Single-cell suspensions were prepared from spleens of Pfs25-VLP-immunized mice and were sorted to identify antigen-specific memory B cells (using the sorting markers CD19$^+$; IgM$^-$; IgD$^-$; CD38$^+$; CD95$^+$, APC-labeled soluble Pfs25) or plasmablasts/plasma single cells (using the sorting markers CD138$^{high}$; IgM$^-$; IgD$^-$; IgA$^-$; CD3$^-$). For each mouse spleen, the following fluorophore-conjugated antibodies were used: 4 µL CD19 (6D5, Pacific Blue, Biolegend, Cat. No. 115523), 3 µL IgM (RMM-1, APC-Cy7, Biolegend, Cat. No. 406516), 3 µL IgD (11-26c.2a, APC-Cy7, Biolegend, Cat. No. 405716), 3 µL CD38 (90, FITC, eBioscience, Cat. No. 11-0381-81), 4 µL CD95 (Jo2, PE, BD Bioscience, Cat. No. 554258). Eighty 96-well plates were sorted from 11 immunized Kymice and antigen-specific B cells were processed by PCR to amplify Ig H- and L-chain sequences for sequencing. Ig sequences were transfected into HEK293 cells (Expi293F™ Cells, Gibco, Cat. No. A14635) for antibody expression as described below.

Atreca's Immune Repertoire Capture® (IRC™) was used to obtain full-length, natively paired, H- and L-chain variable region sequences from single plasmablasts [CD138$^{high}$ (BV420, Biolegend, 142508, 1.3 µg/ml); IgM$^-$ (APC, Biolegend, 406509, 1.3 µg/ml); IgD$^-$ (APC-Cy7, Biolegend, 405716, 1.3 µg/ml); IgA (PE, eBioscience, 12-4204-82, 1.3 µg/ml); CD3$^-$ (FITC, Biolegend, 100204, 3.3 µg/ml)] of 13 Pfs25-VLP immunized (1564 paired H- and L-chain antibody sequences) and 14 empty-VLP immunized (1437 pairs) Kymice. Cell lysis, reverse transcription, PCR, barcode assignment, sequence assembly, V(D)J assignment, and identification of mutations were performed as described previously[40,41] with the following modifications: biotinylated Oligo(dT) was used for reverse transcription, cDNA was extracted using Streptavidin C1 beads (Life Technologies), PCR primers against mouse gamma, kappa, and lambda constant region sequences replaced human constant region primers, DNA concentrations were determined using qPCR (KAPA SYBR® FAST qPCR Kit for Titanium, Kapabiosystems), and a minimum coverage of 10 reads was required from each chain assembly to be included in the sequence repertoires. V(D)J assignment and mutation identification was performed using an implementation of SoDA[42].

Paired H- and L-chain sequences within each Pfs25-VLP-immunized mouse plasmablast repertoire were assigned to the same cluster if the H-chain V-gene, HCDR3 length, L-chain V-gene, and LCDR3 length were identical. H- and L-chain CDRs, as defined[43], were identified by aligning protein sequences to a hidden Markov model[44]. Sequences were further separated into putative lineages based on the degree of identity of the HCDR3 and LCDR3 sequences. Lineages that were also observed as expanded in the plasmablast repertoires of Kymice immunized with

empty VLP preparation were excluded from further consideration. The remaining lineages were scored based on plasmablast cell clone frequency in the origin repertoire (score proportional to abundance), the degree of SHM in the complete H- and L-chain variable regions (score proportional to degree of SHM), and apparent convergent selection across animals. One or two paired H- and L-chain sequences were selected for expression from each of the 93 lineages that received the highest scores. Selected sequences were either from the plasmablast clone in the lineage with the highest identity to the consensus sequence of the lineage or from the clone expressed by the greatest number of plasmablasts in the lineage.

**Expression and purification of Kymouse-derived IgGs**. Memory B cell-derived antibodies were expressed without purification and cloning, with antibody supernatant collected on day 8 after transfection and screened for binding to Pfs25 by homogeneous time-resolved fluorescence (HTRF) and surface plasmon resonance (SPR). 225 of 555 expressed and screened antibodies showed binding by homogeneous time-resolved fluorescence and surface plasmon resonance and Ig H- and L-chain sequences were obtained for 110 Pfs25-binding antibodies.

In addition, plasmablast and memory B cell-derived antibodies were synthesized by LakePharma, Inc. (Belmont, CA). Each gene sequence was cloned into LakePharma's proprietary high-expression mammalian vector. Briefly, variable regions were synthesized and subcloned into expression vectors containing the appropriate IgG1 H- and L-chain constant region sequences. Each construct was transiently transfected into HEK293 cells. The conditioned media from the transient production run was harvested and clarified by centrifugation and filtration. The supernatant was loaded over a Protein A column. The antibody was eluted with a low pH buffer. Purified antibodies were dialyzed against PBS and analyzed using LabChip GXII. Antibodies were screened for binding against Pfs25-VLP, empty VLP, and soluble Pfs25 using an ELISA format assay. Positives for further analysis were identified by binding to Pfs25 or Pfs25-VLP, but not to the empty VLP with a signal of greater than tenfold the background level.

**Expression and purification of Kymouse-derived Fabs**. $V_L$ and $V_H$ regions (sequences available in Supplementary Data 1) were cloned into pFUSE expression vectors (Invivogen) upstream of human Igκ, Igλ, or Igγ1 $C_H$1 constant regions, as appropriate. Fabs were transiently expressed in HEK293F cells (Thermo Fisher Scientific, Cat. No. R79007) and purified via KappaSelect or LambdaSelect affinity chromatography (GE Healthcare), followed by cation exchange chromatography (MonoS, GE Healthcare) and size exclusion chromatography (Superdex 200 Increase 10/300 GL, GE Healthcare).

**BLI-binding studies**. Biolayer interferometry (Octet RED96, FortéBio) experiments at 25 °C were conducted to determine the epitope bins and binding kinetics of selected Fabs for Pfs25. For competition studies, *Nicotiana benthamiana*-produced Pfs25 was diluted into kinetics buffer (PBS, pH 7.4, 0.01% (w/v) BSA, 0.002% (v/v) Tween-20) at 10 μg/mL and immobilized onto Ni-NTA (NTA) biosensors (FortéBio). Following a 30 s baseline step, biosensors were dipped into wells containing the primary antibody (10 μg/mL) for 10 min, followed by another 30 s baseline, and then dipped into wells containing the secondary antibody (10 μg/mL) for 5 min. Similarly, for binding kinetic studies, *Nicotiana benthamiana*-produced Pfs25 was diluted into kinetics buffer and immobilized onto Ni-NTA (NTA) biosensors (FortéBio). Following a 60 s baseline step, biosensors were dipped into wells containing twofold dilution series of Fab or IgG. Sensors were then dipped back into kinetics buffer to monitor the dissociation rate. Competition and kinetics data were analyzed using FortéBio's Data Analysis software 9.0, and kinetic curves were fitted to a 1:1 binding model. Mean kinetic constants reported are the result of three independent experiments.

**Co-crystallization and structure determination**. For crystallization studies, a Pfs25 construct with potential N-linked glycosylation sites at residue positions 91, 144, and 166 mutated to Gln was cloned into the pHLsec vector via AgeI and KpnI sites. Pfs25 was transiently expressed in HEK293F cells and purified via a 5 mL HisTrap FF column (GE Healthcare), followed by size exclusion chromatography (Superdex 200 Increase 10/300 GL, GE Healthcare). Purified Fabs and Pfs25 were mixed in a 1:2 molar ratio and excess Pfs25 was purified away via size exclusion chromatography (Superdex 200 Increase 10/300 GL, GE Healthcare). Fab-Pfs25 complexes were then concentrated to 10 mg/mL and mixed 1:1 with mother liquor and setup in hanging or sitting drop crystallization experiments. 1190-Pfs25 crystals grew in 0.085 M Tris pH 8.5, 0.17 M sodium acetate, 25.5% (w/v) PEG 4000, and were cryoprotected in 15% (v/v) glycerol. 1245-Pfs25 crystals grew in 0.1 M citric acid pH 5.0, 1 M lithium chloride, 20% (w/v) PEG 6000 and were cryoprotected in 20% (v/v) glycerol. 1260-Pfs25 crystals were grown in 2.0 M ammonium sulfate, 0.1 M Tris pH 8.5 and were cryoprotected in 15% (v/v) ethylene glycol. 1262-Pfs25 crystals grew in 0.2 M magnesium acetate and 20% (w/v) PEG 3350 and were cryo-protected with 20% (v/v) glycerol. 1269-Pfs25 crystals grew in 0.2 M di-ammonium hydrogen citrate and 20% (w/v) PEG 3350 after microseeding from thin needle-like crystals that grew in 0.1 M Tris pH 7.0, 0.2 M sodium chloride, 30% (w/v) PEG 3000. 1276-Pfs25 crystals grew in 0.2 M di-ammonium hydrogen phosphate and 18% (w/v) PEG 3350 and were cryoprotected in 10% (v/v) glycerol. Data were collected at the 08ID-1 beamline at the Canadian Light Source (CLS), processed, and scaled using XDS[45]. The structures were determined by molecular replacement using Phaser and using Pvs25 as a search model[46]. Refinement of the structures was carried out using phenix.refine[47] and iterations of refinement using Coot[48]. All software were accessed through SBGrid[49]. No crystals that led to high-resolution structures were obtained for 1202-Pfs25, 1266-Pfs25, 1267-Pfs25, and 1268-Pfs25 complexes.

**SMFA**. Antibodies were buffer exchanged in phosphate-buffered saline (PBS) and tested at multiple concentrations in SMFA (*Anopheles stephensi* (Nijmegen strain)) as previously described[26]. Four independent feeding experiments were conducted. In the first two feeds (Feed #1 and #2), 1245 and 1269 mAbs were tested alone (*n* = 20 mosquitoes per condition). In Feed #3 and #4, the two mAbs were examined either alone (*n* = 20) or as mixtures. A mixture condition of 1245 (31 μg/ml) and 1269 (4 μg/ml) was tested once in Feed #3 (*n* = 20), and the other three conditions (45-7, 73-13, and 132-27 μg/ml for 1245–1269, respectively) were tested in two feeding experiments (*n* = 20 in Feed #3, and *n* = 40 in Feed #4).

**Statistical analysis**. The best estimate and 95% confidence intervals (95% CIs) of %TRA in mean oocyst density, and the *p* value (whether the observed %TRA was significantly different from no inhibition) for each feed or combination of two (Feed #3 and #4) were calculated using a zero-inflated negative binomial model[50]. Using the same model, we evaluated whether a mixture of two antibodies showed higher %TRA (fewer oocysts) than a single antibody, for each mixture condition. For each antibody, the best-fit line (shown in Fig. 7b, c) and $IC_{80}$ values were calculated using all four feeding data shown in Fig. 7a. The theoretical additive effect (Bliss independence model; i.e., assuming the two mAbs act independently) was also determined by the line fits.

**Data availability**. The crystal structures reported in this manuscript have been deposited in the Protein Data Bank, www.rcsb.org (PDB ID codes 6AZZ, 6B08, 6B0A, 6B0E, 6B0G, and 6B0H). The authors declare that all other data supporting the findings of this study are available within the article and its Supplementary Information files, or are available from the authors on request.

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

## Acknowledgements

We thank Christian F. Ockenhouse for a critical review of the manuscript and Wayne Volkmuth for analyses and generation of figures. This work was funded by the PATH Malaria Vaccine Initiative under Grant OPP1108403 from the Bill & Melinda Gates Foundation. X-ray diffraction experiments were performed using beamline 08ID-1 at the Canadian Light Source, which is supported by the Canada Foundation for Innovation, Natural Sciences and Engineering Research Council of Canada, the University of Saskatchewan, the Government of Saskatchewan, Western Economic Diversification Canada, the National Research Council Canada, and the Canadian Institutes of Health Research. We would like to acknowledge the Structural & Biophysical Core Facility, The Hospital for Sick Children, for access to the Octet RED96 BLI instrument. SMFA activities were supported in part by the intramural program of the NIAID, NIH.

## Author contributions

Experimental conception and design: S.W.S., B.M., A.B., K.M., S.C., S.R., F.L., W.R.S., D.E., A.B., C.R.K., J.-P.J.; data acquisition: S.W.S., B.M., A.B., K.M., Q.L., S.C., and S.R.; analysis of data: S.W.S., B.M., A.B., K.M., Q.L., S.C., N.N., E.G., S.R., P.K., F.L., W.R.S., D.E., C.R.K., and J.-P.J.; supplying key reagents: V.Y.; drafting the article or revising it critically for important intellectual content: S.W.S., B.M., K.M., A.B., F.L., D.E., W.R.S., P.K., C.R.K., and J.-P.J.
