## [Peer Review File · Nature Communications]

Reviewers' comments:

Reviewer #1 (Remarks to the Author):

The authors define the crystal structure of 6 distinct, transmission-blocking mAbs bound to recombinant Pfs25. The antibodies were generated in a Kymouse, which contain the human heavy and light chain variable region loci. Consequently the core variable region of the mAB produced is made up of human V, (D) and J exons. The goal of using this mouse line is to facilitate the production of recombinant antibody that could be used in humans. The work confirms the transmission-blocking potential of a site in Pfs25 on EGF-domain 3 previously found to interact with mouse mAB 4B7 and identifies a second site that includes aa from multiple EGF domains. They also carefully compare the sequences and binding specificities of antibodies obtained from different mice and find Ig with similar sequences are induced in distinct Kymice following immunization. They state that this information can be used to enable the structure-guided design of immunogens that might increase the immunogenicity of potent epitopes (lines 89-90), but do not explain how they would do this. Specific strategies/examples should be included in the discussion to demonstrate how the structural information can be used to enhance vaccine development. Site 1 has been known and already used to focus the response (Stowers et al. 2000 and McGuire et al. 2017). Increasing immunogenicity is much more challenging.

Although beyond the scope of the current manuscript, much additional work is needed to understand of the development of the immune response against VPL-Pfs25. Samples through the immunization process are needed to analyze the maturation of the Ig repertoire through the transition from the initial activated B-cell population to memory cell production and a comparison of the SMFA activity over time. Does transmission-blocking activity constantly improve or does it reach a peak and then plateau or decline?

It would also be informative to know what epitopes are bound by the antibodies that do not block transmission. Did any of the non-transmission blocking Ig compete with the transmission-blocking Ig for Pfs25 binding.

Specific comments

The initial 2 Results paragraphs need to be clarified to better describe the Ig populations and the selection of the 10 mAb for further evaluation. Specific suggestions are below.

Lines 106- 109: Two different cell populations are single cell sort from the spleen after the 4th immunization to identify the Ig molecules, plasmablasts and Pfs25-specific memory cells. The rationale for these two approaches is not given and needs to be included.

Line 110: How many of the mAb from the 93 lineages identified in the plasmablast population bound to Pfs25?

Lines 112-115 & fig 1/line 653: Does Fig 1 show most of the plasmablast sequences (1,561 out of the 1,564)? If so, which of these were selected as the 95 mAb in the 93 lineages? The 95 should be indicated on the figure.

Are the sequences that were similar to those in the VPL only immunized mice included?
How many memory cell sequences were included? All 225 that bound Pfs25?

Line 117: How often do distinct Kymice generate similar Ig against a specific immunogen?
In humans it is my understanding that it is rare to have two individuals generate a similar Ig against an immunogen. On the other hand it is not surprising that the Ig sequences of plasmablasts and memory cells isolated after the 4th immunization would have a similar repertoire and I am not clear as to why they highlight this.

Line 120: How many of the 35 mAbs selected for SMFA and the 18 that blocked transmission, where from plasmablasts?

Line 123-124: Table S1 only shows data from 10 mAb.

Lines 130-131. Please elaborate on the potential significance of similar sequences for mAb with transmission-blocking being identified in both the plasmablast and memory cell repertoires.

Line 207: Does this mean that in one Ig the L-chain variable region binds the Pfs25 region bound by the H-chain variable region of another Ig. Please clarify the sentence.

Line 326-327: More specifics about how the ability to adopt multiple conformations can be used to direct immunogen design are need to state that this is critical knowledge? Are there ways to engineer this into a recombinant antigen.

Line 342: Please include a recent publication demonstrating that including the loop region from EGF domain 3 expressed on the Ad virus capsid induced transmission-blocking antibodies, McGuire et al Mal J. 2017.

Line 380- I would indicate that the method used to express antibodies is explained in more detail below in the Methods section titled Recombinant expression and purification of Kymouse-derived IgGs.

Line 346-347: Please delete the final sentence. This work has characterized the response of these mAb to Pfs25, but it has not demonstrated that the use of the Kymouse model to generate mAb derived from human V, (D) and J exons gives more information than prior work in mice with WT Ig loci. Selection is a major component of Ig repertoire formation and until the Pfs25 epitopes of transmission-blocking Ig produced in humans are identified we will not know the value of this approach.

Supplementary Data

Fig S3: Please add the affinity of 1245 in Fig S3

Fig S6: Include description of the orientation of the Pfs25 in Fig S6, perhaps labeling the EFG domains or a few aa positions. Use a brighter color to outline the interaction site and describe it and the color scheme for the electrostatic interactions in the legend

Figs S7 and S8 are quite clear.

Fig S9: In marked contrast Fig S9 does not clearly demonstrate that the antibodies will not bind to the tiled structure. Having the antibodies shown if the binding is blocked is not helpful. I realize it is hard to show a negative result, but it is an important point and it would be worth trying to develop a better picture. Perhaps it would be better to just showing the interaction of the (color coded) 4 domains in the proposed tiled structure, to demonstrate how the EGF domain 3 epitope is buried.

Tables S4-10: Please Include IC50 or affinity of the Ig for Pfs25 in Tables S4-10 to allow direct comparison without having to look back to the table in the paper.

Reviewer #2 (Remarks to the Author):

The manuscript by Scally et al., entitled "Molecular definition of multiple sites of antibody inhibition of malaria transmission-blocking vaccine antigen Pfs25" isolated and characterized Plasmodium falciparum antibodies against antigen Pfs25 elicited by immunization of Pfs25 virus-like particles in human immunoglobulin loci transgenic mice. The authors further reported six crystal structures of Pfs25 in complex with selected antibodies, which revealed detailed contact information between the Pfs25 antigen and the cognate antibodies. The authors showed that these antibodies recognize two distinct immunogenic sites on Pfs25. Using standard membrane feeding assay (SMFA), the authors were able to assay the capacity of the antibodies to inhibit malaria transmission in vivo. Interestingly, the authors found that the transmission inhibition capacity of antibodies can be additive if antibodies targeting distinct Pfs25 epitopes were combined in the assay.

This study is well designed, with conclusions fully supported by data. The strength of the manuscript is summarized as below, which will be valuable for the field for basic understanding of malaria protective immunity and future vaccine design. Some minor points are also listed for improvement.

Strength

The structure information of Plasmodium falciparum transmission-blocking antigen Pfs25 is not available. The structure of a homolog protein Pvs25 derived from Plasmodium vivax has been used in the past as the structural reference for Pfs25. This study reveals the structure of Pfs25 and the structural distinction between Pfs25 and Pvs25 will fill the knowledge gap of the field.

This study uncovered the molecular basis of the recognition between malaria transmission-blocking antibodies elicited by Pfs25 VLP particle in human Ig repertoire knock-in transgenic mice and the transmission-blocking antigen Pfs25. Such information will be very useful for structure-based malaria transmission-blocking vaccine design.

The characterization of the genetics and biochemical features of malaria transmission-blocking antibodies provides in depth understanding of the protective mechanisms, and the evolution of the protective humoral immunity.

Minor points

1). Figs. 7e and f seem to be swapped. In Fig 7e, YL52 is shown in the image, while S52F is listed in the figure legend. In Fig. 7f, FL52 is shown in the image while S52Y is listed in the figure legend.

2). Please provide the manufacturer information for the adjuvant, the clone name and source of the antibody conjugates for FACS sorting.

Reviewer #3 (Remarks to the Author):

This manuscript describes obtaining and analyzing human antibodies against the Pfs25 protein of the malarial parasite. Six crystal structures of complexes between Pfs25 and each of the six anti-Pfs25 Fab fragments are presented. Each antibody inhibits the growth of parasites in a mosquito that takes a blood meal containing the antibody.

The manuscript is plagued with unclear descriptions. The authors examine many antibodies, perhaps 1564 antibodies, but "down-select" 95 from the 1564. What is "down-select"? Another example is Line 115, "This analysis revealed the breadth of the B cell response (Fig. 1)". Does this mean an extensive breadth or a surprisingly small breadth. Please say which it is.

Technically, the manuscript appears to be done well. This reviewer thinks though that there isn't much new and unique data here. Many antibodies have been encountered by the authors. Many antibodies have been shown to bind to Pfs25. Fewer antibodies have been shown to have transmission blocking activity. Then six crystal structures are shown, which locate the epitopes of the antibodies, but not much more.

Figs 1 and 3 should be removed from the manuscript. The results shown in Fig. 1 should be said in the text and may be less confusing there. The results of Fig. 3 are all similar and this figure should go into the supplement as well.

Lesser points:

Line 71, "tested extensively in human trials". Citation?

Fig. 1, this reviewer does not see the importance of this large figure. Readers are not told how this figure relates to obtaining antibodies or immunity against Pfs25. Why show the antibodies identified as to the individual mouse that they came from? Please justify the inclusion of this figure or remove it from the manuscript. It could go to the supplement.

This reviewer doubts that much immunogen engineering can be done to EGF-like domains, as they are small and are cross-linked by disulfide bonds.

Line 66, awkward wording. "...immunization strategies protecting from several Pf life stages." Protecting from? Protecting humans from? Protecting against?

Pvs25, 46% sequence identity. Of which are 22 conserved cysteines in disulfide bonds. Not surprising that the Pfs25 structure was very similar to the earlier vivax one.

Figs. S1, S2, Please describe in the legends what the conclusions of the figures are. This reviewer thinks that the bottom rightmost panels show the results of immunization, but it took time and thought to work that out. Describe what each panel means, so that the reader can focus on the results.

Lines 130-131, "...providing examples of similar B-cell diversity being present in both immune repertoires for SMFA-active mAbs". Please tell the reader why finding similar B-cell diversity in both immune repertoires is important. Nature Communications is a general-reader journal.

225 out of 555 Ab were confirmed for binding to Pfs25 by SPR.

Line 157, "Generally, antibody recognition of site 2 was associated with slower on-rates and slower off-rates." But there are only two site-2 antibodies. Please don't use the word "Generally".

Fig. 5 and 6, How is this center-of-mass and angle-of-approach analysis useful?

Fig. 7 legend, define "SHM" again here.

Line 237, HDCR3

Fig. 7, This is an interesting look at the effects of somatic hypermutation.

Response to Reviewers' comments:

Reviewer #1 (Remarks to the Author):

The authors define the crystal structure of 6 distinct, transmission-blocking mAbs bound to recombinant Pfs25. The antibodies were generated in a Kymouse, which contain the human heavy and light chain variable region loci. Consequently the core variable region of the mAB produced is made up of human V, (D) and J exons. The goal of using this mouse line is to facilitate the production of recombinant antibody that could be used in humans. The work confirms the transmission-blocking potential of a site in Pfs25 on EGF-domain 3 previously found to interact with mouse mAB 4B7 and identifies a second site that includes aa from multiple EGF domains. They also carefully compare the sequences and binding specificities of antibodies obtained from different mice and find Ig with similar sequences are induced in distinct Kymice following immunization. They state that this information can be used to enable the structure-guided design of immunogens that might increase the immunogenicity of potent epitopes (lines 89-90), but do not explain how they would do this. Specific strategies/examples should be included in the discussion to demonstrate how the structural information can be used to enhance vaccine development. Site 1 has been known and already used to focus the response (Stowers et al. 2000 and McGuire et al. 2017). Increasing immunogenicity is much more challenging.

We agree that increasing the immunogenicity of Pfs25 is the main challenge associated with this TBV antigen. In the Discussion, we provide several examples of how our structural findings will guide the design of next-generation immunogens that might increase the immunogenicity of potent epitopes. For example, on p.16: "Our six crystal structures revealed that the loop regions of the EGF-like domains are not rigid; instead, they are capable of adopting multiple conformations, providing important knowledge for immunogen design seeking to elicit B-cell responses against the most potent Pfs25 conformation." Furthermore, also on p. 16: "Nevertheless, our data [...] highlight the benefits of polyclonal responses that can act additively." And finally, on p.17: "An understanding of the Pfs25 three-dimensional structure and the location of protective epitopes will also enable the optimal display and density of Pfs25 on nanoparticles, which is known to significantly boost B-cell responses." We also note that the work to explore these ideas, and others, is ongoing and beyond the scope of the present study.

Although beyond the scope of the current manuscript, much additional work is needed to understand of the development of the immune response against VPL-Pfs25. Samples through the immunization process are needed to analyze the maturation of the Ig repertoire through the transition from the initial activated B-cell population to memory cell production and a comparison of the SMFA activity over time. Does transmission-blocking activity constantly improve or does it reach a peak and then plateau or decline?

We agree with the reviewer that understanding the development of the SMFA activity over time is an important question that will be the subject of future work.

It would also be informative to know what epitopes are bound by the antibodies that do not block transmission. Did any of the non-transmission blocking Ig compete with the transmission-blocking Ig for Pfs25 binding.

We agree with the reviewer that understanding the molecular details of competition between non-transmission blocking and transmission blocking antibodies would be highly informative for vaccine

design. Unfortunately, our study does not provide such insight as we focused on only active mAbs.

Specific comments

The initial 2 Results paragraphs need to be clarified to better describe the Ig populations and the selection of the 10 mAb for further evaluation. Specific suggestions are below.

Lines 106- 109: Two different cell populations are single cell sort from the spleen after the 4th immunization to identify the Ig molecules, plasmablasts and Pfs25-specific memory cells. The rationale for these two approaches is not given and needs to be included.

Sorting both B cell populations allows an assessment of how representative the antibody producing plasma cells (directly derived from plasmablast) and the source of the serum polyclonal response reflect the antigen experienced memory cell population. A good vaccine should have a degree of equivalence but as yet the mechanism for equal or skewed population of memory and plasma cell populations are not well understood. Therefore, the rationale of the present study was to determine whether identical antibody lineages were seen in the memory and plasmablast populations. Our study shows that both compartments contain functional antibodies. To clarify these points, we added the following sentence on pp.5-6: “The anti-Pfs25 response in both cell types was assessed to determine which compartment contains the most functionally potent, active, or broad set of antibodies.” In the Discussion on p.16, we added the sentence “Our data shows that both memory B cell- and plasmablast-derived mAbs contain functional antibodies. It is therefore likely they both contribute to the serum polyclonal response (from short and long term plasma cells) and therefore may also be part of a long-lived response, which is an important goal in the development of a durable TBV.”

Line 110: How many of the mAb from the 93 lineages identified in the plasmablast population bound to Pfs25?

In the Results section on p.6, we now explicitly provide this information: “1564 natively paired, complete variable region sequences were derived from plasmablasts.” And later “95 plasmablast-derived mAbs, representing 93 unique, putative lineages, and 18 memory B cell-derived mAbs were generated after down-selection using sequence analyses and Ig lineage clustering as criteria (described in Methods). 34 of the 95 plasmablast-derived mAbs were confirmed to bind to Pfs25-VLP and not empty VLP by ELISA (Supplementary Fig. 3C). Only 7 plasmablast-derived mAbs bound to soluble Pfs25 indicating most required avidity for binding, or Pfs25 on VLPs has epitopes not present in soluble Pfs25 (Supplementary Fig. 3C).” Supplementary Fig. 3C has also been added during revision to clarify this point. This figure provides a summary of Pfs25 ELISA results for plasmablast-derived and memory B cell-derived sequences according to their Ig lineage and the number of cells per lineage.

Lines 112-115 & fig 1/line 653: Does Fig 1 show most of the plasmablast sequences (1,561 out of the 1,564)? If so, which of these were selected as the 95 mAb in the 93 lineages? The 95 should be indicated on the figure.

Are the sequences that were similar to those in the VPL only immunized mice included?

How many memory cell sequences were included? All 225 that bound Pfs25?

There are 1564 PB natively-paired, full-length variable regions and 18 MBC natively-paired, full-length variable regions in the tree of Fig. 1. The 95 PB clones selected for recombinant expression and assays are now noted in the figure, as are the 18 MBC clones that were recombinantly expressed and tested. All 1564 plasmablast sequences from Pfs25-VLP immunized Kymice and 18 sequences from the

memory B cell screen are included in the tree of 1582 natively-paired, full-length variable regions from 1582 individual cells. The subset of plasmablast sequences from Pfs25-VLP immunized Kymice that are similar to plasmablast sequences from empty VLP immunized Kymice are included as part of the 1564 plasmablast dataset shown in the tree. This presentation shows the immune response in the plasmablast compartment to Pfs25-VLP, the complete immunogen used. This information has been added to the Figure 1 legend to improve clarity.

Many selection criteria were used to down-select antibodies for recombinant expression, including removing sequences similar to those obtained from the empty VLP immunizations. These details are now provided in more detail in the Methods section on pp.19-20: “Lineages that were also observed as expanded in the plasmablast repertoires of Kymice immunized with empty VLP preparation were excluded from further consideration. The remaining lineages were scored based on plasmablast cell clone frequency in the origin repertoire (score proportional to abundance), the degree of SHM in the complete H- and L-chain variable regions (score proportional to degree of SHM), and apparent convergent selection across animals. One or two paired H- and L-chain sequences were selected for expression from each of the 93 lineages that received the highest scores. Selected sequences were either from the plasmablast clone in the lineage with the highest identity to the consensus sequence of the lineage or from the clone expressed by the greatest number of plasmablasts in the lineage.”

Line 117: How often do distinct Kymice generate similar Ig against a specific immunogen? In humans it is my understanding that it is rare to have two individuals generate a similar Ig against an immunogen. On the other hand it is not surprising that the Ig sequences of plasmablasts and memory cells isolated after the 4th immunization would have a similar repertoire and I am not clear as to why they highlight this.

In the Results section on p.6, we now explicitly provide this information “This analysis revealed a diverse response, spanning 784 Ig lineages across 6 VH gene families (Fig. 1). Interestingly, anti-Pfs25 antibodies of highly similar sequence were observed in different individual Kymice, with 102 Ig common lineages being present in two or more similarly immunized Kymice (Supplementary Fig. 3D).” Supplementary Fig. 3D has also been added during revision to clarify this point. This figure provides a summary of Pfs25 ELISA results for plasmablast-derived and memory B cell-derived sequences according to the number of instances an Ig lineage is observed in one or more mice.

Line 120: How many of the 35 mAbs selected for SMFA and the 18 that blocked transmission, where from plasmablasts?

On p.6, we have added the following language to clarify this information: “When a subset of 35 mAb binders were tested for functional activity in the membrane feeding assay at a single high concentration of 375 µg/ml (or lower, as one antibody had limiting material), 19 showed inhibition in a single concentration functional screen above 80% and one greater than 50%. These 20 mAbs were derived from 15 independent lineages with three only found in plasmablasts, six derived from lineages in both plasmablasts and memory B-cells, and six only from memory B-cells. Ten of these antibodies were selected for further analysis (Table 1).”

Line 123-124: Table S1 only shows data from 10 mAb.

We thank the reviewer for pointing out this omission. Table S1 has now been updated to include SMFA data for all 35 mAbs tested.

Lines 130-131. Please elaborate on the potential significance of similar sequences for mAb with transmission-blocking being identified in both the plasmablast and memory cell repertoires.

As noted above, we now provide additional language to better explain the rationale behind our approach and the potential significance of our findings. In the Discussion on p.16, we added the sentence “Our data shows that both memory B cell- and plasmablast-derived mAbs contain functional antibodies. It is therefore likely they both contribute to the serum polyclonal response (from short and long term plasma cells) and therefore may also be part of a long-lived response, which is an important goal in the development of a durable TBV.”

Line 207: Does this mean that in one Ig the L-chain variable region binds the Pfs25 region bound by the H-chain variable region of another Ig. Please clarify the sentence.

The reviewer correctly interpreted the point we were trying to make. We have now modified this sentence to improve its clarity: “Despite a rotation of $\sim 173^\circ$ compared to 1190/1276, mAb 1262 only shifts its center of mass by 3.5° effectively swapping its variable L- and H-chain domains with the H- and L-chains of 1190/1276 (Fig. 4B).”

Line 326-327: More specifics about how the ability to adopt multiple conformations can be used to direct immunogen design are need to state that this is critical knowledge? Are there ways to engineer this into a recombinant antigen.

We have work ongoing to test whether specific loop conformations on Pfs25 can be rigidified to improve immunogenicity and elicit more potent antibody responses. These studies are beyond the scope of the present report. We have now adjusted our claim that our structural findings provide “important knowledge” to direct immunogen design (p.16).

Line 342: Please include a recent publication demonstrating that including the loop region from EGF domain 3 expressed on the Ad virus capsid induced transmission-blocking antibodies, McGuire et al Mal J. 2017.

As suggested by the reviewer, we now include this reference on p.16.

Line 380- I would indicate that the method used to express antibodies is explained in more detail below in the Methods section titled Recombinant expression and purification of Kymouse-derived IgGs.

We have now indicated that the methods used to express antibodies is described below.

Line 346-347: Please delete the final sentence. This work has characterized the response of these mAb to Pfs25, but it has not demonstrated that the use of the Kymouse model to generate mAb derived from human V, (D) and J exons gives more information than prior work in mice with WT Ig loci. Selection is a major component of Ig repertoire formation and until the Pfs25 epitopes of transmission-blocking Ig produced in humans are identified we will not know the value of this approach.

As requested by the reviewer, we have deleted this sentence.

Supplementary Data

Fig S3: Please add the affinity of 1245 in Fig S3

In Fig. S3, all memory B cell-derived antibodies for which SPR was performed are labeled. 1245 is a plasmablast-derived antibody and therefore wasn't measured in SPR assays, as described in the Methods. There is no primary screening binding data for memory B cell-derived 1276, 1260 and 1266 antibodies because of the way the high-throughput assays were performed. We have added a sentence in the Fig. S3 legend to highlight this fact: "The binding affinity (nM) of 225 Pfs25 antigen binding antibodies assayed by SPR to surface immobilized Pfs25 antigen with the positions of antibodies 1262, 1267, 1268 and 1269 indicated in blue. Antibodies 1276, 1260 and 1266 were not expressed at sufficient quantity in this high-throughput assay format to allow SPR measurements."

Fig S6: Include description of the orientation of the Pfs25 in Fig S6, perhaps labeling the EFG domains or a few aa positions. Use a brighter color to outline the interaction site and describe it and the color scheme for the electrostatic interactions in the legend

As requested by the reviewer, we have now labeled the EGF-like domains, brightened the interaction site outlines in the figure, and provided a description in the figure legend.

Figs S7 and S8 are quite clear.

Fig S9: In marked contrast Fig S9 does not clearly demonstrate that the antibodies will not bind to the tiled structure. Having the antibodies shown if the binding is blocked is not helpful. I realize it is hard to show a negative result, but it is an important point and it would be worth trying to develop a better picture. Perhaps it would be better to just showing the interaction of the (color coded) 4 domains in the proposed tiled structure, to demonstrate how the EGF domain 3 epitope is buried.

As requested by the reviewer, we have altered this figure to improve its clarity. As suggested, we have now removed the Abs from the figure, colored the central Pvs25 molecule according to EGF-like domain and outlined site 1a contacted Pfs25 residues to demonstrate that they are buried in this tile-like packing.

Tables S4-10: Please Include IC50 or affinity of the Ig for Pfs25 in Tables S4-10 to allow direct comparison without having to look back to the table in the paper.

As requested by the reviewer, we have now included the K_D of the antibody for Pfs25 in Tables S4-10 to allow direct comparison.

Reviewer #2 (Remarks to the Author):

The manuscript by Scally et al., entitled "Molecular definition of multiple sites of antibody inhibition of malaria transmission-blocking vaccine antigen Pfs25" isolated and characterized Plasmodium falciparum antibodies against antigen Pfs25 elicited by immunization of Pfs25 virus-like particles in human immunoglobulin loci transgenic mice. The authors further reported six crystal structures of Pfs25 in complex with selected antibodies, which revealed detailed contact information between the Pfs25 antigen and the cognate antibodies. The authors showed that these antibodies recognize two distinct immunogenic sites on Pfs25. Using standard membrane feeding assay (SMFA), the authors were able to assay the capacity of the antibodies to inhibit malaria transmission in vivo. Interestingly, the authors found that the transmission inhibition capacity of antibodies can be additive if antibodies targeting distinct Pfs25 epitopes were combined in the assay.

This study is well designed, with conclusions fully supported by data. The strength of the manuscript is summarized as below, which will be valuable for the field for basic understanding of malaria protective immunity and future vaccine design. Some minor points are also listed for improvement.

Strength

The structure information of Plasmodium falciparum transmission-blocking antigen Pfs25 is not available. The structure of a homolog protein Pvs25 derived from Plasmodium vivax has been used in the past as the structural reference for Pfs25. This study reveals the structure of Pfs25 and the structural distinction between Pfs25 and Pvs25 will fill the knowledge gap of the field.

This study uncovered the molecular basis of the recognition between malaria transmission-blocking antibodies elicited by Pfs25 VLP particle in human Ig repertoire knock-in transgenic mice and the transmission-blocking antigen Pfs25. Such information will be very useful for structure-based malaria transmission-blocking vaccine design.

The characterization of the genetics and biochemical features of malaria transmission-blocking antibodies provides in depth understanding of the protective mechanisms, and the evolution of the protective humoral immunity.

We thank the reviewer for these encouraging comments.

Minor points

1). Figs. 7e and f seem to be swapped. In Fig 7e, YL52 is shown in the image, while S52F is listed in the figure legend. In Fig. 7f, FL52 is shown in the image while S52Y is listed in the figure legend.

We thank the reviewer for noticing this error, which has now been rectified.

2). Please provide the manufacturer information for the adjuvant, the clone name and source of the antibody conjugates for FACS sorting.

We have now added the requested information in the Material and Methods section. On p.18: “The immunization regimen consisted of three IP immunizations performed on days 0, 21, 42 using 1 µg of Pfs25-VLP formulated in Montanide ISA720 (70% v/v, Seppic Inc), and a final boost at day 72 with 1 µg of unadjuvanted Pfs25-VLP, also administered IP.” And further: “The following fluorophore conjugated antibodies were used: CD19 (6D5, Pacific Blue, Biolegend, Cat Num 115523), IgM (RMM-1, APC-Cy7, Biolegend, Cat Num 406516), IgD (11-26c.2a, APC-Cy7, Biolegend, Cat Num 405716), CD38 (90, FITC, eBioscience, Cat Num 11-0381-81), CD95 (Jo2, PE, BD Bioscience, Cat Num 554258).”

Reviewer #3 (Remarks to the Author):

This manuscript describes obtaining and analyzing human antibodies against the Pfs25 protein of the malarial parasite. Six crystal structures of complexes between Pfs25 and each of the six anti-Pfs25 Fab fragments are presented. Each antibody inhibits the growth of parasites in a mosquito that takes a blood meal containing the antibody.

The manuscript is plagued with unclear descriptions. The authors examine many antibodies, perhaps 1564 antibodies, but “down-select” 95 from the 1564. What is “down-select”?

The path of steps that led from the sequencing data to the 95 plasmablast paired sequences selected for recombinant expression and testing (referred to as “down-selection” in the text) is now presented

in additional detail in the Materials and Methods section (pp.19-20): “Paired H- and L-chain sequences within each Pfs25-VLP immunized mouse plasmablast repertoire were assigned to the same cluster if the H-chain V-gene, HCDR3 length, L-chain V gene, and LCDR3 length were identical. H- and L-chain CDRs, as defined, were identified by aligning protein sequences to a hidden Markov model. Sequences were further separated into putative lineages based on the degree of identity of the HCDR3 and LCDR3 sequences. Lineages that were also observed as expanded in the plasmablast repertoires of Kymice immunized with empty VLP preparation were excluded from further consideration. The remaining lineages were scored based on plasmablast cell clone frequency in the origin repertoire (score proportional to abundance), the degree of SHM in the complete H- and L-chain variable regions (score proportional to degree of SHM), and apparent convergent selection across animals. One or two paired H- and L-chain sequences were selected for expression from each of the 93 lineages that received the highest scores. Selected sequences were either from the plasmablast clone in the lineage with the highest identity to the consensus sequence of the lineage or from the clone expressed by the greatest number of plasmablasts in the lineage.” On p.6, we have modified the text to explicitly call-out the Methods section when we introduce our down-selection approach: “95 plasmablast-derived mAbs, representing 93 unique, putative lineages, and 18 memory B cell-derived mAbs were generated after down-selection using sequence analyses and Ig lineage clustering as criteria (described in Methods).”

Another example is Line 115, “This analysis revealed the breadth of the B cell response (Fig. 1)”. Does this mean an extensive breadth or a surprisingly small breadth. Please say which it is.

As requested by the reviewer we have clarified this sentence and provided additional information on p6: “This analysis revealed a diverse response, spanning 784 Ig lineages across 6 VH gene families (Fig. 1).”

Technically, the manuscript appears to be done well. This reviewer thinks though that there isn’t much new and unique data here. Many antibodies have been encountered by the authors. Many antibodies have been shown to bind to Pfs25. Fewer antibodies have been shown to have transmission blocking activity. Then six crystal structures are shown, which locate the epitopes of the antibodies, but not much more.

Figs 1 and 3 should be removed from the manuscript. The results shown in Fig. 1 should be said in the text and may be less confusing there. The results of Fig. 3 are all similar and this figure should go into the supplement as well.

As suggested by the reviewer, Fig. 3 has now been moved to the supplementary section (now Supplementary Figure 4). Fig. 1 has been altered to improve its clarity and make its content more accessible to the reader. Namely, we now clearly indicate the VH gene usage of IgG sequences in the tree, and highlight all plasmablast and memory B cells that were selected for further characterization to investigate the wide-ranging genetic diversity of the anti-Pfs25 B cell response in Kymice. As described above in response to Reviewer 1, we now also provide additional analyses related to Fig. 1 that are shown in Supplementary Figure 3.

Lesser points:

Line 71, “tested extensively in human trials”. Citation?

We now include two citations that describe Pfs25 immunogen human trials.: 1) Wu, Y. *et al.* Phase 1

trial of malaria transmission blocking vaccine candidates Pfs25 and Pvs25 formulated with montanide ISA 51. *PLoS ONE* 3, e2636 (2008). 2) Talaat, K. R. *et al.* Safety and Immunogenicity of Pfs25-EPA/Alhydrogel®, a Transmission Blocking Vaccine against *Plasmodium falciparum*: An Open Label Study in Malaria Naïve Adults. *PLoS ONE* 11, e0163144 (2016).

Fig. 1, this reviewer does not see the importance of this large figure. Readers are not told how this figure relates to obtaining antibodies or immunity against Pfs25. Why show the antibodies identified as to the individual mouse that they came from? Please justify the inclusion of this figure or remove it from the manuscript. It could go to the supplement.

As noted above in response to other comments, we have significantly modified the content of Fig. 1 to make its content more informative to the reader: we now clearly indicate the VH gene usage of each IgG sequence in the tree, highlighting the wide-ranging diversity of the anti-Pfs-VLP B cell response; we also label all plasmablast and memory B cell antibody sequence clones selected for further characterization to better highlight the “down-selection” process; we also provide additional analyses related to Fig. 1 that are shown in Supplementary Figure 3: C) binding reactivity of Ig lineages and D) similarity of B cell responses between Kymice. We believe this new presentation make Fig. 1 valuable for the molecular understanding of multiple sites of antibody inhibition of malaria transmission-blocking vaccine antigen Pfs25.

This reviewer doubts that much immunogen engineering can be done to EGF-like domains, as they are small and are cross-linked by disulfide bonds.

We agree with the reviewer that stabilization of the core structure of Pfs25 is unlikely to be required in immunogen engineering efforts because of the stability of its EGF-like fold as mediated through extensive disulfide bonds. However, as noted in the Discussion on p.16: “Our six crystal structures revealed that the loop regions of the EGF-like domains are not rigid; instead, they are capable of adopting multiple conformations, providing important knowledge for immunogen design seeking to elicit B-cell responses against the most potent Pfs25 conformation.” As such, we believe that immunogen engineering can be explored for sub-regions of Pfs25. These studies are ongoing and are beyond the scope of the present work.

Line 66, awkward wording. “...immunization strategies protecting from several Pf life stages.” Protecting from? Protecting humans from? Protecting against?

We have now changed the sentence to “immunization strategies effective at blocking several *Pf* life stages.”

Pvs25, 46% sequence identity. Of which are 22 conserved cysteines in disulfide bonds. Not surprising that the Pfs25 structure was very similar to the earlier vivax one.

We agree with the reviewer that the similarity between the Pvs25 and Pfs25 structures is not surprising. As stated in the Results section on p.9: “Comparison of the structures of Pfs25 in each complex to the previously determined Pvs25 structure (Supplementary Fig. 6A)¹³ showed that Pfs25 adopts a similar overall structure as Pvs25, in which four distinct EGF-like domains assume a triangular arrangement and disulfide bonding patterns are conserved (Supplementary Fig. 6B).” Nonetheless, in this paragraph we also highlight that: “Differences between the six Pfs25 structures and Pvs25 (Supplementary Fig. 6D) were considerably greater (overall RMSD= 2.1 Å), particularly in the EGF-like

domain 4 (RMSD= 4.0 Å, Supplementary Fig. 6C)."

Figs. S1, S2, Please describe in the legends what the conclusions of the figures are. This reviewer thinks that the bottom rightmost panels show the results of immunization, but it took time and thought to work that out. Describe what each panel means, so that the reader can focus on the results.

Details have been added to the Figs. S1 and S2 legends to better communicate the results to the reader:

Fig. S1. Serum polyclonal titers measured by ELISA for Kymice engineered to express the full set of human immunoglobulin variable, diversity and joining region gene segments for the immunoglobulin heavy chain and the kappa light chain variable and joining region gene segments (HK Kymice). Serum polyclonal antibody titer for each immunized mouse (KMFC numbers) were determined by limiting dilution ELISA against plate bound VLP-Pfs25 (a and d), VLP (VLP-Empty) (b and e) and soluble Pfs25 protein (c and f). Kymice were immunized with either naked VLP (VLP-Empty (panels a to c) or VLP-Pfs25 (panels d to f). All mice raised a polyclonal antibody response to the VLP regardless of the presence or absence of the additional Pfs25 antigen (b and e), but only VLP-Pfs25 mice raised a polyclonal response to Pfs25 (Panel a and d compared to panels c and f).

Fig. S2. Serum polyclonal titers measured by ELISA for Kymice engineered to express the full set of human immunoglobulin variable, diversity and joining region gene segments for the immunoglobulin heavy chain and the lambda light chain variable and joining region gene segments (HL Kymice). Serum polyclonal antibody titer for each immunized mouse (KMFC numbers) were determined by limiting dilution ELISA against plate bound VLP-Pfs25 (a and d), VLP (VLP-Empty) (b and e) and soluble Pfs25 protein (c and f). Kymice were immunized with either naked VLP (VLP-Empty (panels a to c) or VLP-Pfs25 (panels d to f). All mice raised a polyclonal antibody response to the VLP regardless of the presence or absence of the additional Pfs25 antigen (b and e), but only VLP-Pfs25 mice raised a polyclonal response to Pfs25 (Panel a and d compared to panels c and f).

Lines 130-131, "...providing examples of similar B-cell diversity being present in both immune repertoires for SMFA-active mAbs". Please tell the reader why finding similar B-cell diversity in both immune repertoires is important. Nature Communications is a general-reader journal.

Both compartments are sources for diversity of antibodies that are immunogen-specific. Immunogen-specific antibodies are critical factors in successful humoral immune responses. As stated above in response to Reviewer 1, we added the following sentence on pp.5-6 to clarify these points: "The anti-Pfs25 response in both cell types was assessed to determine which compartment contains the most functionally potent, active, or broad set of antibodies." In the Discussion on p.16, we added the sentence "Our data shows that both memory B cell- and plasmablast-derived mAbs contain functional antibodies. It is therefore likely they both contribute to the serum polyclonal response (from short and long term plasma cells) and therefore may also be part of a long-lived response, which is an important goal in the development of a durable TBV."

225 out of 555 Ab were confirmed for binding to Pfs25 by SPR.

This is correct. We have re-worded this section to improve clarity on p.6: "555 memory B cell-derived mAbs were generated as human IgG1 and screened for binding to Pfs25 by homogeneous time-

resolved fluorescence and surface plasmon resonance. 225 mAbs were confirmed by both assays to bind Pfs25, and had affinities ranging from ~500 nM to less than 1 nM (Supplementary Fig. 3B)."

Line 157, "Generally, antibody recognition of site 2 was associated with slower on-rates and slower off-rates." But there are only two site-2 antibodies. Please don't use the word "Generally".

The word "generally" has been removed from the sentence.

Fig. 5 and 6, How is this center-of-mass and angle-of-approach analysis useful?

The angle of approach is a measure used to describe and compare how the Abs recognize Pfs25. On p.11, we note that our highest-affinity binder against Pfs25 site 1, 1269, "demonstrates an angular difference of 31° and 35° in its angle of approach compared to 1190/1276 and 1262, respectively, [...] (Fig 4B and F, Supplementary Table 7)." Similar analyses in HIV have described angles of approaches of mAbs that have informed vaccine design (Tran *et al.*, PNAS 2014; Zhou *et al.*, Cell 2015).

Fig. 7 legend, define "SHM" again here.

The definition of SHM has been added in the figure legend.

Line 237, HDCR3

We thank the reviewer for noticing this typo. It has now been fixed to HCDR3.

Fig. 7, This is an interesting look at the effects of somatic hypermutation.

We thank the reviewer for this encouraging comment.

REVIEWERS' COMMENTS:

Reviewer #1 (Remarks to the Author):

The Ig selection process is much clearer, except line 129, Where did the 35 binders tested by SMFA come from? If you include the 7 from plasmablasts and 18 from memory cells that is only 25. If you include all 34 plasmablast Ig with the 18 from memory cells that is 52?

Other than that all revisions have been made.

Reviewer #3 (Remarks to the Author):

All concerns have been addressed.

Response to Reviewers' comments:

Reviewer #1 (Remarks to the Author):

The Ig selection process is much clearer, except line 129, Where did the 35 binders tested by SMFA come from? If you include the 7 from plasmablasts and 18 from memory cells that is only 25. If you include all 34 plasmablast Ig with the 18 from memory cells that is 52?

As requested by the reviewer we have clarified this sentence on p6: "A subset of 35 mAb derived from 20 plasmablast and 15 memory B cell mAbs were selected for further functional activity in the membrane feeding assay."

Other than that all revisions have been made.

Reviewer #3 (Remarks to the Author):

All concerns have been addressed.

We thank the reviewer for this positive assessment of our revised manuscript.